

# State-of-the-art artificial intelligence approaches for anomaly detection and remaining useful life prediction: a review

Mohd Khidir Gazali[1], Khairunnisa Hasikin[1,2], Khin Wee Lai[1], Aizat Hilmi Zamzam[3] and Rafat Damseh[4]

[1] Department of Biomedical Engineering, Faculty of Engineering, Universiti Malaya, Kuala Lumpur, Wilayah Persekutuan Kuala Lumpur, Malaysia
[2] Center of Intelligent Systems for Emerging Technology (CISET), Faculty of Engineering, Universiti Malaya, Kuala Lumpur, Wilayah Persekutuan Kuala Lumpur, Malaysia
[3] Engineering Services Division, Ministry of Health, Putrajaya, Wilayah Persekutuan Putrajaya, Malaysia
[4] Department of Computer Science and Software Engineering, United Arab Emirates University, Al Ain, Abu Dhabi, United Arab Emirates

Corresponding author
Khairunnisa Hasikin,
khairunnisa@um.edu.my

## ABSTRACT

**Background:** Accurate prediction of the remaining useful life (RUL) of assets is fundamental to the development of effective maintenance strategies and overall asset management. Despite significant advancements, there remains a notable gap in integrating fault detection and diagnostics (FDD) with RUL prediction models to create more comprehensive and accurate maintenance systems. One of the key challenges in this field is the limited ability of current models to generalize effectively across different types of equipment and varying operating conditions. This gap emphasizes the need for further research and innovation in developing robust and adaptable RUL prediction methodologies that can be applied broadly across diverse industrial scenarios.

**Methodology:** This review systematically evaluates the machine learning (ML) and deep learning (DL) techniques used for anomaly detection and RUL prediction, focusing on their efficacy and practical application. By adhering to the Preferred Reporting Items for Systematic Review and Meta-Analyses (PRISMA) criteria, the review identifies and addresses the deficiencies in existing models. It explores a range of machine learning and deep learning methods, including probabilistic approaches, hybrid models that combine multiple machine learning techniques, and neural networks designed to handle large-scale time-series data. The review also examines the potential for synergy between machine learning models and FDD, aiming to enhance the precision of equipment monitoring and the early detection of defects. The challenges of data variability, the irregularity in equipment deterioration, and the interpretability of complex models are highlighted.

**Results:** The analysis reveals that while current machine learning and deep learning models have made considerable strides in predicting the RUL of assets, significant challenges remain, particularly in their ability to generalize across various equipment types and operational contexts. Hybrid models and neural networks have shown promise in improving the accuracy of RUL predictions, especially when managing large, complex datasets. However, the irregular nature of equipment wears and tear, coupled with data variability, continues to pose significant challenges. The review highlights the need for more robust and adaptable models that can not only predict

RUL more accurately but also integrate seamlessly with FDD systems to provide a more holistic approach to maintenance.

**Conclusion:** This comprehensive review focusses on the need for continued research in developing more integrated, generalizable, and efficient predictive maintenance systems. By exploring the application of AI in virtual assistants, the review suggests promising avenues for extending asset longevity and optimizing maintenance schedules. While current models offer valuable insights, they must evolve to address the identified gaps in generalizability and model interpretability.

# INTRODUCTION

Remaining useful life (RUL) is an estimate of the amount of time a machinery or system is likely to operate before it requires repair or replacement. It is a critical component in predictive maintenance strategies, allowing for timely decision-making to prevent failures and optimize operational efficiency (*Seman et al., 2023*; *Wang et al., 2022*). RUL predictions can be derived from various approaches, including physics-based, statistical-based, data-driven or hybrid which combine two or more method together to forecast the lifespan of equipment (*Wang et al., 2022*). When the RUL reaches zero, it signifies that the system or equipment has attained its failure point, indicating the end of its operational life. At this stage, it can no longer fulfill its intended function effectively or safely (*Arunthavanathan et al., 2023*; *Zhang et al., 2021*).

In machine learning (ML), supervised learning methods and neural networks are commonly employed because of their capacity to manage extensive time-series data and extract pertinent features automatically for precise RUL prediction (*Wu, Ding & Huang, 2020*). *Martins, Vale & Maitelli (2015)* use conventional neural network (CNN) with an adaptive shrinkage processing mechanism to predict the operational lifespan of machinery before it requires maintenance or to be replaced. Long short-term memory (LSTM) been used to detect early failures in rotating machinery by analyzing vibration data and learning to identify fault patterns (*Lee & Chang, 2020*). These deep learning models are very effective at identifying complicated patterns in time-series data. In addition, fault detection and diagnosis (FDD) were used in engineering to identify, localize and often diagnose faults or abnormalities in systems. The FDD bases can be categorized in four different types; Model, data-driven, knowledge and statistic and hybrid based approaches (*Ozkat et al., 2023*; *Zhao et al., 2019*). *Martins, Vale & Maitelli (2015)* use model-based together with ML-based approaches, developed for fault detection and isolation (FDI), with studies exploring the combination of both to enhance FDI performance. A hybrid-based method combining multivariate empirical mode decomposition, fuzzy entropy and an optimized support vector machine (SVM) for wind energy converter fault diagnosis, achieving high

diagnostic accuracy under varying conditions (*Zhang et al., 2023c*). *Soualhi et al. (2022)* and *Fong et al. (2023)* use fault detection sensor in chiller plants, employing a hybrid algorithm that integrates ML with pattern recognition for effective fault diagnosis.

Despite notable progress in the estimation of RUL and FDD, there exist various research insufficiencies that hinder the optimal integration of these methodologies into holistic maintenance plans (*Liao & Tian, 2012*). While deep learning models generally produce very accurate findings, their complexity and lack of interpretability might make it difficult to acquire trust and approval from maintenance personnel. Similarly, classic machine learning approaches can have interpretability issues, making it more difficult to understand the reasoning behind forecasts (*Hu et al., 2023*). There remains an insufficiency of research to concentrate integration with all four main topics identified earlier to address the knowledge gap. Such interest like factor contributes the extending of RUL from anomaly detection with artificial intelligence was far from complete. The potential to considerably extend the lifespan of equipment through strategies that go beyond traditional maintenance, such as Artificial Intelligence (AI) and FDD, is still a subject of ongoing exploration and development. This discipline has not yet reached maturity, as there are still a multitude of challenges and uncertainties that must be resolved.

A foundation for future research and innovation in this critical domain is established by systematic reviews, which are instrumental in identifying these knowledge gaps and advancing our understanding. While existing review articles on anomaly detection and RUL prediction offer valuable insights, they present specific limitations that our article addresses. For example, the published review by *Zhang et al. (2023a)* concentrates exclusively on methodologies that are utilized in mechanical systems, without investigating broader industrial applications or integration with FDD. In the same aspect, *Kumar et al. (2024)* investigates rotating machinery techniques, but it does not provide any coverage of advanced AI-based methods or their generalizability beyond this field. In a different review, the practical integration of these techniques with FDD systems is not addressed, despite the fact that it emphasizes deep learning approaches (*Serradilla et al., 2022*). Moreover, *Ferreira & Gonçalves (2022)* emphasizes the practical applications of machine learning, but it fails to address the hybrid approaches and real-world obstacles associated with integrating these methods into FDD applications. Conversely, our article addresses these deficiencies by integrating RUL prediction with FDD, thereby encompassing a diverse array of industries and advocating for the practical application of hybrid and AI-driven strategies. Our research addresses real-world operational challenges and offers actionable insights for both academia and industry by emphasizing the synergy between physical and data-driven models. Our review is positioned as a substantial contribution that complements and expands the existing *corpus* of literature as a result of this comprehensive perspective.

In light of the challenges and opportunities discussed, this review explores the latest technologies revolutionizing predictive maintenance, with a focus on the critical concept of RUL. By exploring with these methodologies, the review seeks to bridges the gap between foundational principles of RUL estimation and the innovative, data-driven strategies that

are transforming the field. This exploration not only highlights the technical advancements but also emphasizes their practical implications, aiming to provide actionable insights for industry professional, educators and policymakers. Finally, our effort aims to influence current maintenance practices, set new standards for industrial operations, and drive the creation of curricula that include the most recent advances in ML and predictive maintenance. In addition, it promotes a better knowledge of RUL and its critical role in improving efficiency, sustainability, and competitiveness across sectors.

## MOTIVATION AND SIGNIFICANCE OF STUDY

This study is addressing several key gaps identified from the existing literature. Inadequate study on integration between anomaly detection and RUL estimation. However, as we looked through the current research, we noticed a few important things missing as summarized in Table 1. One of the biggest gaps is that most studies look at anomaly detection and RUL prediction as separate problems. The review conducted by *Zhang et al. (2023a)* offers a robust methodological framework for RUL estimation in mechanical systems, yet confines its analysis to particular sectors, such as rotating machinery. It neglects to consider broader industrial applicability or the practical integration with FDD systems, which are paramount for effective real-world implementation. Similarly, *Kumar et al. (2024)* stresses the value of signal centric and statistical techniques for rotating elements; however, it overlooks the integration of modern artificial intelligence frameworks, such as deep learning, transfer learning, or hybrid designs, that are progressively significant in current industrial applications. Very few try to bring them together into a single, connected approach yet doing so could lead to much more accurate and responsive maintenance strategies. Without addressing their synergistic potential when integrated. The absence of comparative evaluation across several industrial fields, there is limited literature evaluating cross industry for generalizability and adaptability of ML models for asset health monitoring. There is also a lot of potential in hybrid models that combine physical understanding of how machines fail with powerful data-driven AI methods. These can offer the best of both worlds: they are grounded in science, but flexible enough to handle real-world noise and complexity. But despite this promise, we found that these approaches haven't been explored as much as they should be.

The review conducted by *Rana (2025)* concentrates on the predictive maintenance and fault detection in electrical power systems that are driven by AI. Although it exhibits promising developments, its purview is limited to smart grids and does not account for the integration with RUL estimation or cross-sector adaptability, which are both critical for generalizable asset health monitoring. Similarly, *Han et al. (2024)* do not discuss unified frameworks that combine AD with RUL in dynamic industrial contexts, despite their comprehensive survey of fault diagnosis under varying operational conditions. *Neupane et al. (2025)* provide a thorough examination of machine learning for the detection of machinery faults. Nevertheless, they devote insufficient attention to hybrid modelling methods and the difficulties associated with the implementation of integrated systems across multiple domains.

**Table 1 Overview of key review articles highlighting gaps in anomaly detection, fault diagnosis, and RUL estimation.**

| References | Focus area | Methodologies | Industrial domain | Key limitations/Gaps |
|---|---|---|---|---|
| Zhang et al. (2023a) | RUL estimation frameworks | Traditional ML methods | Rotating machinery | Lacks integration with FDD systems; sector-specific analysis |
| Kumar et al. (2024) | Signal-based and statistical techniques for diagnostics | Classical statistical methods | Rotating elements | Ignores modern AI (deep learning, transfer learning); lacks hybrid integration |
| Rana (2025) | Predictive maintenance and fault detection | AI for smart grid diagnostics | Electrical power systems | Limited to smart grids; does not integrate with RUL or generalize across sectors |
| Han et al. (2024) | Fault diagnosis under dynamic conditions | Fault diagnosis survey | General industrial systems | No discussion of unified AD-RUL frameworks |
| Neupane et al. (2025) | ML for machinery fault detection | Machine learning | Multiple sectors (generalized) | Limited coverage of hybrid modeling; lacks implementation strategies |
| Serradilla et al. (2022) | Deep learning for prognostics | Deep learning models | Manufacturing systems | Overlooks DL limitations (interpretability, data dependency); lacks FDD integration |
| Ferreira & Gonçalves (2022) | ML applications in various sectors | General ML approaches | Cross-sectoral | Superficial coverage of hybrid models; ignores imbalanced/scarce data challenges |

Furthermore, there is a lack of comparative evaluations across various industries. There is still a lack of research on the adaptability and generalizability of machine learning models for asset health surveillance. Hybrid models that integrate the adaptability of data-driven AI methods with the physical principles of machine degradation demonstrate significant potential. These models maintain the adaptability necessary to manage chaotic, imbalanced, or limited failure data, while also providing the scientific rigour of traditional methods. However, these hybrid strategies are rarely investigated in a unified and cross-industry context, despite their potential.

In addition, the review by Serradilla et al. (2022) accentuates deep learning models, particularly in the realm of prognostics for manufacturing systems. Nevertheless, it does not critically evaluate the limitations associated with deep learning concerning data dependency, interpretability, or integration with fault detection frameworks. Concurrently, Ferreira & Gonçalves (2022) provide a practical overview of machine learning applications across diverse sectors; however, they lack a comprehensive analysis of hybrid models that amalgamate physical and data-driven methodologies, and they do not confront the challenges posed by imbalanced, noisy, or scarce failure data.

The exhaustive and integrative focus on the convergence of three critical components in predictive maintenance anomaly detection (AD), RUL estimation, and FDD distinguishes this study from previous reviews. In contrast to previous research, which has tended to investigate these components in isolation or with minimal overlap, this review systematically addresses their synergistic integration, which is crucial for the development of practicable and reliable predictive maintenance frameworks. This study not only categorizes advanced AI approaches including deep learning, ensemble methods, and hybrid models but also critically analyzes how these can be effectively integrated with FDD

systems and applied in real-world industrial contexts, in contrast to earlier reviews that primarily concentrate on specific methodologies (*e.g.*, deep learning or signal-based analysis) or narrow industrial sectors (such as rotating machinery or smart grids). Additionally, this review broadens the scope by assessing the cross-sector adaptability of these integrated approaches across a variety of domains, such as aerospace, energy, manufacturing, and medical equipment. This comprehensive applicability directly addresses a critical limitation in the existing literature, which frequently lacks generalizability and overlooks the complexities inherent in diverse operational contexts. By offering a holistic perspective that bridges methodological advancements with authentic deployment scenarios, this research fills a significant gap and lays the groundwork for the next generation of intelligent, industry agnostic maintenance systems capable of adapting across heterogeneous implementation ecosystems.

# METHODOLOGY

## Search terms identification

The concept of asset life cycle encompasses a broad spectrum of keywords that are crucial for understanding its behavior throughout its expected lifespan. Selecting the right keyword is essential to find appropriate academic articles, as keywords serve as the primary tools for indexing and retrieving research articles in databases. To reflect the main objectives of this review, the keywords were chosen based on four main topics: Equipment Lifespan Prediction, FDD, Prognostics and Maintenance Management, and Machine Learning (Table 2). The selected keywords collectively embody the essence of a article, allowing for the targeted identification of relevant studies without the need to filter through irrelevant material. This includes a wide range of approaches and perspectives, allowing for a thorough examination of the junction of FDD, RUL prediction, maintenance management, and AI.

## Research questions

In this study, we created a table of research questions to address significant topics and gaps discovered in the field of RUL, FDD and predictive maintenance (Table 3). By addressing these research questions, the study aims to contribute and improve understanding the concepts and applicable models for equipment reliability and maintenance.

## Identification for reporting guideline

Selecting an appropriate research framework before screening research article is a first step in the systematic literature review process, thereby ensuring only the most relevant and high-quality studies are selected. The Preferred Reporting Items for Systematic Reviews and Meta-Analyses (PRISMA) technique was selected because of its suitability to the method applied in these studies and also widely adopted by high-impact journals and institutions (*Oumaima, Benabdellah & Zellou, 2023*). Thus, it facilitating the peer-review and publication processes (*Sewell, Schellinger & Bloss, 2023*). PRISMA employs a structured methodology that includes the following steps: first, the identification of pertinent studies through the use of predefined keywords, followed by the screening of

**Table 2 Research main category and its common keyword.**

| Topic | Keywords |
|---|---|
| Equipment life span prediction | Remaining useful life, life cycle analysis, life span, survival analysis, weibull distribution, reliability engineering, circular economy, life data analysis and time-to-failure analysis |
| Fault detection and diagnosis (FDD) | Fault detection, anomaly detection, sensor data analysis, pattern recognition, performance degradation analysis, condition monitoring, root cause analysis and fault tolerance system |
| Prognostic and maintenance management | Health management, preventive maintenance, corrective maintenance, prognostic/predictive maintenance, work order management, asset management, maintenance strategy, health index, reliability-centered maintenance, total productive maintenance, prognostic health management, |
| Artificial intelligence (AI) | Machine learning, deep learning, neural networks, natural language processing, cognitive computing and artificial intelligence |

**Table 3 Table of research questions.**

| No. | Research questions |
|---|---|
| RQ1 | What are the range of studies associated with predicting the RUL of equipment? |
| RQ2 | How can hybrid models combining physical and data-driven methods improve RUL prediction for complex systems? |
| RQ3 | What are the comparative accuracies between Artificial Intelligence approaches in RUL prediction? |
| RQ4 | How effective are FDD approaches in detecting early indicators of equipment failure? |
| RQ5 | What challenges are associated with the implementation of sensor technology for fault detection? |
| RQ6 | What are the most effective methods use for monitoring condition in the predictive maintenance and how can root cause analysis be effectively determined in fault tolerance systems? |
| RQ7 | How do life cycle analysis and survival analysis frameworks influence the selection of maintenance strategies and health indices in reliability driven maintenance? |

articles to eliminate duplicates and irrelevant research, the assessment of the eligibility of studies based on specific criteria, the systematic collection and organization of data, and the final visualization of the findings through research maps and summaries (*Zamzam et al., 2021*). This process, which is facilitated by a 27-item checklist and 16 sub-items, ensures that only the most relevant and reliable studies are included, thereby facilitating reproducibility and peer review (*Page et al., 2021*) and. It also helps maintain high quality and transparency in the review.

## Search strategy and data collection

The selection process prioritizes studies that employ machine learning techniques for prognostic anomaly detection and predictive maintenance. Using only the most relevant studies to enhances the quality data and focusing on each research findings (*PRISMA 2020 Checklist, 2020*). The inclusion and exclusion criteria as tabulated in studies were meticulously selected to ensure that the review concentrates on pertinent, high-quality studies that are consistent with its objectives (Table 4). In order to guarantee current, accessible, and original contributions, only research articles published in English between 2010 and 2024 were considered. Non-peer-reviewed sources, including conference articles, case studies, book chapters, and guidelines, were excluded due to their infrequent use of comprehensive data analysis, standardized peer-review processes, or detailed methodologies. To prevent the potential for language barriers that could influence the

**Table 4 Inclusion and exclusion studies.**

| Inclusion and exclusion criterion | | |
| --- | --- | --- |
| Criterion | Inclusion | Exclusion |
| Sources | Research article | Review article, conference article, proceedings article, case study, chapter in book, book section, encyclopedia, early access, guideline and other sources |
| Language | English | Non-English |
| Period | Between 2010 to 2024 | Before 2010 |
| Selection journal | (1) Focus on the using of method to find remaining useful life for all kind of equipment at various ages and condition or (2) Focus on the use of machine learning algorithm in prognosis either for faulty detection or predictive maintenance | Other than related equipment remaining useful life, prognosis, predictive maintenance and machine learning algorithm |

interpretation of technical content, non-English studies were excluded. Additionally, publications prior to 2010 were omitted in order to emphasize more recent developments in the field. In order to maintain relevance and assure alignment with the review's scope, articles that were unrelated to predictive maintenance, FDD, machine learning in prognostics, or RUL were excluded. These criteria establish a precise framework for the identification of studies that contribute to the advancement of knowledge in this critical field while simultaneously preserving rigor and focus. The keywords were then strategically combined into a single search string using Boolean operators (*e.g.*, AND, OR) to ensure a comprehensive yet targeted search across multiple databases (Table 4). This approach enhances the search process by including all relevant studies while minimizing irrelevant results, providing a robust foundation for the systematic review. Data was subtracted from eight major databases; ScienceDirect, Scopus, IEEE Xplore, Web of Science, Emerald, MEDLINE Complete, Dimensions, and Springer Link from year 2010 until 2024. Google Scholar was excluded to avoid duplicate articles. Articles are initially selected based on relevance keywords and then screened for quality.

## Quality assessment

A total 27 checklist item is divided into five main categories; preparation data, methods to cover RUL criteria for study selection, data collection, bias assessment, and results findings (*Oumaima, Benabdellah & Zellou, 2023*; *Page et al., 2021*). Figure 1 and Table 5 provide an overview of the search results at different stages of the screening process. A total of 27,507 journal articles were retrieved from eight major databases. After removing 652 duplicate records and 10,410 records under exclusion criteria, 16,445 records were screened. From these, 10,678 records were excluded, leaving 5,767 articles for further screening. Subsequently, 786 reports were sought for retrieval, with 522 reports not retrieved. This left 220 reports for eligibility assessment. During this assessment, 85 reports were excluded for not being related to engineering (50), asset management (23), or prediction of asset behavior (12). Ultimately, 22 studies were included in the review, emphasizing the importance of evaluating the quality of each study to mitigate the risk of bias.

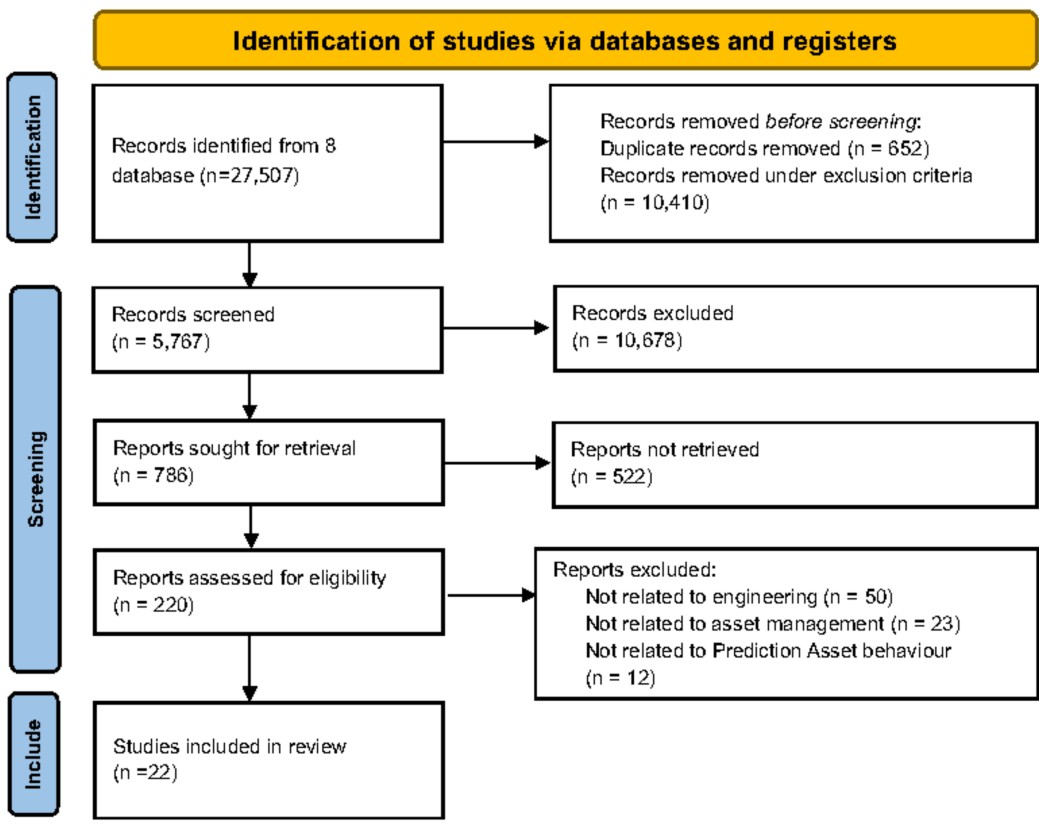

**Figure 1 PRISMA reporting guideline identification, screening, and inclusion process from 27,507 to 22 selected articles.** The diagram is a PRISMA flowchart. It describes the process from identification, screening, and final selection studies in a systematic review from an initial pool of records.

**Table 5 RUL Boolean keyword search result across eight major databases.**

| Search strings | Science Direct | Scopus | IEEE Explore | WoS | Emerald | Medline complete | Dimensions | Springer Link |
|---|---|---|---|---|---|---|---|---|
| ("Remaining useful life") AND ("Fault detection" OR "Anomaly detection") AND ("Machine Learning") | 731 | 31 | 44 | 31 | 25 | 327 | 6,344 | 268 |
| ("Remaining Useful life") AND ("prognostic health management" OR "Predictive Maintenance") AND ("Machine learning") | 717 | 99 | 53 | 95 | 37 | 169 | 3,796 | 216 |
| ("Machine Learning") AND ("health management" OR "asset management" OR "maintenance strategy") AND ("Remaining Useful Life") | 1,099 | 126 | 231 | 113 | 45 | 245 | 6,549 | 300 |
| ("Machine Learning") AND ("Health Index" OR "Predictive Maintenance") AND ("Remaining useful life") | 814 | 101 | 69 | 100 | 39 | 192 | 4,253 | 247 |
| Total including duplicate | 3,361 | 357 | 397 | 340 | 146 | 933 | 20,942 | 1,031 |
| Subtotal including duplicate | 27,507 | | | | | | | |
| Total selected area after quality assessment | 22 | | | | | | | |

# RESULT

## Main findings

Table 6 presents a summary of 22 research articles on RUL, selected using the PRISMA framework. These studies span various industries and systematically categorized into condition based, asset specific, risk-based, and maintenance-based approaches. Among these, condition-based maintenance combined with predictive analytics emerged as the most prominent approach in the reviewed literature, with over 60% of the studies focusing on this topic. For example, *Aydemir & Acar (2020)* introduced anomaly-triggered RUL estimation method to improve detection during health operation anomalies in aerospace and manufacturing. However, the model's generalizability across a variety of operating environments is restricted by its dependence on a singular anomaly detection method, the Cumulative Sum Control Chart (CUSUM), which is a statistical tool used for monitoring changes in processes over time.

*Wang et al. (2018)* use nonlinear model with first-time hitting detection approach to predict degradation levels under imperfect maintenance in heavy industries. Although this method offers higher accuracy, it heavily depends on stochastic process assumptions, making it unsuitable for broader applications. Similarly *Zhang et al. (2020)* proposed a novel iterative standby system lifetime (SSL) estimation method that integrates both operational and storage degradation processes, offering comprehensive lifetime prediction for manufacturing and aerospace sectors. Although innovative, its iterative approach is not suitable for non-linear degradation models.

Another major topic involves deep learning applications for predictive maintenance. *Zheng, Liao & Zhu (2023)* developed a two-stage model using Robust-ResNet for fault detection and RUL prediction, providing improved classification accuracy across four degradation stages. However, accelerated life testing data used in their study may not fully represent natural degradation patterns. Similarly, *Cheng et al. (2021)* implemented a transferable convolutional neural network (TCNN) for RUL prediction in bearings, showcasing advancements in feature extraction and transfer learning across multiple failure behaviors In aerospace engineering. *Ture et al. (2024)* introduced a stacking ensemble learning method for deep learning-based anomaly detection, leveraging multiple regression algorithms to enhance predictive robustness.

In the context of data-driven approaches, several studies highlighted innovations in health monitoring and feature extraction. *de Pater & Mitici (2023)* expand health indicator functions with similarity based matching methods to predict unhealthy stages of aircraft engines with minimal failure data. *Rosero, Silva & Ribeiro (2022)* presented a novel classification methodology using HI to segregate health degradation into two stages, improving failure detection in aerospace systems. Similarly, *Duan et al. (2023)* applied principal component analysis (PCA) for dimensionality reduction while integrating similarity metrics to construct health indicators and monitor degradation trends effectively. However, the limited exploration of similarity metrics suggests room for further improvement. *Arunthavanathan et al. (2023)* established a self-learning fault detection framework using one-class support vector machines (OC-SVM) and neural network-based

**Table 6 Existing literature on RUL from various field of industry.**

| Author (Region) | Dataset | Topic | | Sector | Predictive method | Novelty | Gap and research opportunities |
|---|---|---|---|---|---|---|---|
| | | Condition base | Asset specific | | | | |
| Aydemir & Acar (2020) Turkey | Public data | ✓ | | Aircraft engine | Anomaly triggered RUL estimation | Address ill define RUL during health operation | Applicability to varying operating conditions only focuses using the CUSUM control chart for anomaly detection. |
| Wang et al. (2018) China | Private data | ✓ | | Draught fan | Nonlinear model and first hitting time to determine degradation level | RUL prediction systems under imperfect maintenance | Model Rely heavily on stochastic process assumption and not universally applicable to all types of industrial equipment |
| de Pater & Mitici (2023) Netherland | Public data | | ✓ | Aircraft engine | Health indicator development and similarity base matching during unhealthy stages | Health state division equipment with few failure | Reliance on the limitation for failure data. Due to preventive maintenance, there are very few labeled samples available for training |
| Zheng, Liao & Zhu (2023) China | Private & public data | ✓ | | Internal Gear Pump | Robust-ResNet, faulty detection and RUL in Four Stages Classifiers | Two-stage deep learning model for fault detection and RUL | Accelerated life testing to gather data, may not represent natural degradation patterns |
| Zhang et al. (2020) China | Private data | ✓ | | Logistic and supply chain, storage parts | General iterative method for standby system lifetime (SSL) estimation, | Combine storage degradation and operational degradation processes spare parts in the lifetime estimation and standby systems | Extending iterative method to non-linear degradation model |
| Pei et al. (2019) China | Private data | ✓ | | Gyroscope | Constructs a nonlinear degradation model | Degradation model considering bivariate time scales | Gap in research on adaptive estimation methods on multiple model and parameters |
| Liu et al. (2020) China | Private data | | ✓ | Marine engineering (Subsea Valve actuator) | Dynamic Bayesian networks, physical models (Fatigue Crack Growth and Corrosion model) | Address structural degradation underwater structure and stress distribution. | Further refine fatigue crack growth models in extreme environmental conditions and material heterogeneity |
| Cheng et al. (2021) China | Private data | ✓ | | Rolling bearings | Convolutional neural network on feature extraction, transfer learning, performance matrices | RUL prediction bearings under multiple failure behaviors | Present models struggle with inconsistent feature distributions caused by different bearing failure modes, impact the generalization ability of the models |
| Ture et al. (2024) Turkey | Public data | ✓ | | Turbofan engine | Anomalies detection using deep learning method, data-driven approach | RUL prediction, Stacking Ensemble Learning and integrates the strengths of multiple regression algorithms | The model does not explicitly account for external factors like environmental conditions or variable operational scenarios that could impact RUL |
| Rosero, Silva & Ribeiro (2022) Portugal | Private data | ✓ | | Aircraft cooling system | Health indicator for failure pattern identification | Novel classification methodology to segregate between two health degradation stages | Adaptive HI prediction by further extending study method evolving asset condition and operating environment. HI base fault detection and isolation classification matrix |
| Arunthavanathan et al. (2023) Canada | Private data | ✓ | | Process industries, manufacturing | One-class support vector machine, neural network permutation algorithm | Self-learning fault detected capability | Fault to failure transition predominantly assume linear degradation model |

(Continued)

| Author (Region) | Dataset | Topic | | Sector | Predictive method | Novelty | Gap and research opportunities |
|---|---|---|---|---|---|---|---|
| | | Condition base | Asset specific | | | | |
| Carroll et al. (2019) UK | Private data | ✓ | | Wind turbine | Supervisory control and data acquisition (SCADA), high-frequency vibration data | Integration of SCADA and high-frequency vibration data for failure prediction. Focus on specific failure mode to achieve tailored prediction | Due to complexity of continuous prediction, The RUL is not estimated as a precise number of hours, days, or months. Instead, it is predicted within predefined time windows. |
| Biondi, Sand & Harjunkoski (2017) Germany | Private data | | ✓ | Processing equipment | Mixed integer linear programming (MILP), state task network (STN) representation | Integrated maintenance and production scheduling problem using RUL as a singular indicator on health asset | Multiple feature extraction techniques need, particularly for complex and noisy data |
| Sayyad et al. (2021) India | Private data | ✓ | | Milling machine | Multi sensor fusion, denoising, signal transformation and predictive model | Systematic integration of AI techniques with multi-sensor data fusion on real-time data | Develop frameworks for effective multi-sensor data integration, noise reduction for data processing |
| Wang & Mamo (2019) Taiwan | Public data | ✓ | | Ball bearing of rotating machinery | ML model (Support Vector regression, Random Forest), Anomaly detection and optimization technique | Introduce novel element for confidence interval using Jackknife method | Dataset variability from a controlled environment with predefined operational conditions. This limits its applicability to real-world scenario |
| Lee, Kim & Lee (2023) South Korea | Private data | ✓ | | Forklift | Multi-faceted methodology combining advanced sensor data analysis and ML Model | Comprehensive Feature Engineering for Forklift Failure Prediction, Data Augmentation Technique | Imbalanced datasets from left, center, and right weight conditions may lead to biased predictions |
| Duan et al. (2023) China, USA | Public data | ✓ | | Aircraft engine | Principal Component Analysis (PCA) and Similarity method | Integrates PCA for dimensionality reduction and similarity-based methods (Manhattan distance) for health indicator | Limitation to exploration of distance metrics for similarity calculation, use PCA to construct health indicator to represent degradation trends |
| Ma, Xu & Yang (2023) Malaysia, China | Private data | ✓ | | Medical equipment | Fault diagnosis, categorization and prediction, experimental validation, ABC optimization algorithm | Life cycle management framework of early, middle and late phases symptom | The model requires extensive data on historical failures and performance, which may not be readily available for all types of equipment |
| Solís-Martín, Galán-Páez & Borrego-Díaz (2023) Spain | Private data public data | ✓ | | Turbojet engine, fast charging battery, bearing | XAI method, matrix of evaluation, quantitative proxies | Introduced quantitative metric to measure time-dependence in explainable AI in evaluating explainability for time-series regression tasks | Limited research available in PHM highlights the need for tailored methods to improve model interpretability and applicability |
| Fan, Nowaczyk & Rögnvaldsson (2020) Sweden | Public data | ✓ | | Turbojet engine | Transfer learning | Develop COSMO for RUL prediction in dynamic and variable conditions | To enhance transfer learning for unseen operating conditions with COSMO model to handle complex data |
| Wang & Zhao (2023) UK | Public data | | ✓ | Turbojet engine | Prediction for multiple condition of complex machinery, three stage feature selection | Groups sensor data from various operating conditions using k-medoids clustering to uncover patterns related to a specific operational state that traditional methods might miss | Effective feature selection for reducing model complexity |
| Liao & Tian (2012) USA, Canada | Private data | ✓ | | Ball beating of rotating machinery | Bayesian approach, accelerated degradation testing (ADT) | Handling time-varying conditions and adaptability to nonlinear model | Advanced degradation modeling, handling more complex operating conditions, incorporating with human factors |

permutation algorithms, offering automated detection for fault progression. *Wang & Mamo (2019)* combined support vector regression and random forest models to introduce confidence intervals using the jackknife method, strengthening anomaly detection in machinery. While innovative, the reliance on controlled environment data limits real-world applicability. Similarly, *Wang & Zhao (2023)* presented a three-stage feature selection method with k-medoids clustering, uncovering operational patterns for complex machinery systems, yet their framework remains complex for large-scale applications. *Sayyad et al. (2021)* applied multi-sensor fusion techniques integrated with denoising and signal transformation to provide robust real-time predictive models for manufacturing applications. However, their framework requires further exploration for effective noise reduction and data integration.

Statistical and optimization models were also explored across multiple studies. *Liao & Tian (2012)* applied Bayesian approaches in combination with accelerated degradation testing (ADT) to handle time-varying conditions and improve adaptability in manufacturing and transportation sectors. Meanwhile, *Carroll et al. (2019)* integrate SCADA data with high-frequency vibration signals for predictive analysis of renewable energy systems, enhancing fault detection in wind turbines. *Biondi, Sand & Harjunkoski (2017)* introduced mixed-integer linear programming (MILP) and state task network (STN) models, offering a structured approach to handling noisy and complex datasets in renewable energy. *Ma, Xu & Yang (2023)* designed the Fine Life Cycle Prediction System for Failure of Medical Equipment to predict failures in medical devices in a structured way. The system comprises the Life Cycle Management Module, Status Detection, Fault Diagnosis and Fault Prediction Module. The module is distinguished by its integration with AI method to facilitate proactive maintenance approaches in healthcare.

*Solís-Martín, Galán-Páez & Borrego-Díaz (2023)* explores the predictive maintenance (PdM) application, predominantly on the Explanation AI (XAI) for prognostics and health management (PHM). The algorithm was applied to Grad-CAM for time-series regression by introducing time and feature dependencies addresses regression, which is inherently harder to explain than classification tasks. *Fan, Nowaczyk & Rögnvaldsson (2020)* and *Wang & Zhao (2023)* has enhanced PdM frameworks for complex machinery by introducing advanced methodologies, the former utilizes a feature representation-based transfer learning (TL) approach with consensus self-organizing models (COSMO) to address maintenance planning and operational issues in turbofan systems, while the latter proposes a three-stage feature selection framework combined with DL models to improve accuracy prediction under variable operating conditions. Meanwhile, *Liao & Tian (2012)* address prediction of single units under time-varying operating conditions using an advanced Bayesian updating methodology for RUL, which accommodates both linear and nonlinear degradation-stress dynamics, providing solutions for stochastic operational scenarios to facilitate real-time assessments while mitigate uncertainties in varying conditions.

The reviewed literature highlighted several challenges, with a major issue being the scarcity of failure data due to preventive maintenance practices, as noted by *de Pater & Mitici (2023)* and *Ma, Xu & Yang (2023)* has discover training models difficulty and

requires better techniques to generate or augment data. *Liu et al. (2020)* and *Ture et al. (2024)* show that absence of environmental factors in models such as extreme conditions and operational variability are often ignored limits model accuracy in actual scenarios. Scaling models to new industries or unseen data remains difficult. *Fan, Nowaczyk & Rögnvaldsson (2020)* and *Cheng et al. (2021)* highlighted transfer learning methods struggle to adapt to different operating conditions or large datasets, reducing their effectiveness. Many models still rely on linear degradation assumptions, which oversimplify real-world trends. *Pei et al. (2019)*, *Zhang et al. (2020)* and *Arunthavanathan et al. (2023)* argue for adaptive approaches that can handle non-linear and multi-phase degradation more effectively. Imbalanced and noisy datasets impact model performance. Studies such as *Biondi, Sand & Harjunkoski (2017)*, *Wang & Mamo (2019)* and *Lee, Kim & Lee (2023)* show that biased predictions occur when data is incomplete or unbalanced.

## Overview of frequently used datasets in RUL and fault diagnosis studies

In this review, the datasets analyzed are classified as publicly available, which are accessible for academic and research purposes, or private and proprietary datasets, which are typically collected in-house and often specific to industrial settings. The quality and characteristics of these datasets play a critical role in influencing model training, validation, and performance evaluation.

Table 7 presents an exhaustive summary of publicly accessible datasets that are frequently employed in research pertinent to condition-based RUL forecasting. These datasets serve as the cornerstone for the development, training, and benchmarking of data-driven prognostic models across a multitude of engineering domains. The table encompasses datasets from various sectors, including aerospace, battery systems, and rotating machinery. For example, the NASA C-MAPSS dataset, which simulates the degradation of turbofan engines, is often utilized in aerospace RUL investigations and comprises measurements such as pressure, temperature, fan speed, and fuel-air ratio across an array of fault scenarios. The N-CMAPSS dataset enhances this with more intricate degradation trajectories and multivariate conditions, rendering it particularly suitable for deep learning and transfer learning frameworks. Likewise, the NASA Battery Usage dataset and the MIT Battery Degradation dataset (*Severson et al., 2019*) are employed to examine battery cycle longevity by documenting charge/discharge patterns, voltage, current, temperature, and internal resistance. Within the domain of rotating machinery, datasets such as the IMS Bearing Dataset, FEMTO-ST, and PRONOSTIA provide high-resolution vibration data amassed under controlled experimental conditions and are extensively utilized for bearing fault detection and degradation analysis. Similarly with Gearbox Dataset provided from PHM on year 2009 provides vibration and temperature data for analyzing degradation on rotating gearbox.

Each dataset delineated in table is accompanied by its respective access link, thereby facilitating other researchers in replicating or extending previous experiments. Nonetheless, certain limitations are acknowledged. A considerable number of the datasets are generated through accelerated life testing or controlled laboratory conditions, which

**Table 7 Publicly available physical dataset for condition base RUL.**

| Public data | Description | Category | Dataset | Measurement parameter | Accessible link |
|---|---|---|---|---|---|
| NASA jet engine simulated data | Commercial modular aero-propulsion system simulation (C-MAPSS) | Turbofan engine degradation simulation | Simulated data combination operation and fault mode | Temperature, pressure, fan speed, coolant bleed, fuel-air ratio | https://data.nasa.gov/dataset/c-mapss-aircraft-engine-simulator-data |
| NASA jet engine simulated data | Commercial modular aero-propulsion system simulation (C-MAPSS) | Turbofan engine degradation simulation | Run to failure trajectories | Fuel flow, fan speed, temperature, pressure, fan flow | https://www.nasa.gov/content/prognostics-center-of-excellence-data-set-repository |
| PHM gearbox dataset | Generic industrial gearbox (Year 2009) | Rotating machinery | Gearbox degradation under realistic operating conditions | Vibration, temperature, speed, torque | https://phmsociety.org/public-data-sets/ |
| NASA randomized battery usage | Battery usage | Battery systems | State of health (SOH) | Voltage, current, temperature over charge/discharge cycles | https://www.nasa.gov/content/prognostics-center-of-excellence-data-set-repository |
| MIT battery degradation dataset | Fast-charging battery dataset (Severson et al.) | Batteries systems | Fast charge durability | SOC, internal resistance, temperature, charge profile | https://data.matr.io/1/projects/5c48dd2bc625d700019f3204 |
| FEMTO bearing | PRONOSTIA | Rotating machine—bearings | Run to failure bearings | Vibration, accelerometer, bearing degradation | https://www.nasa.gov/intelligent-systems-division/discovery-and-systems-health/pcoe/pcoe-data-set-repository/ |

may not entirely capture the intricacies of real-world operational environments. Studies such as those by *Ma, Xu & Yang (2023)* and *de Pater & Mitici (2023)* acknowledge the limitations imposed by insufficient failure data particularly due to widespread preventive maintenance practices that prevent systems from running to failure. Additionally, they frequently lack contextual features such as ambient temperature variations, user variability, or unstructured anomalies. Furthermore, naturally degraded datasets those acquired from actual equipment over extended periods are notably scarce. This limitation undermines the external validity of models trained exclusively on synthetic or laboratory-derived data. Consequently, while these datasets are indispensable for methodological progress, there is a growing exigency for more comprehensive and diverse datasets that more accurately reflect operational uncertainties and realistic degradation behaviors.

## Distribution of publication by component and years of study

This VOSviewer map illustrates the interconnected themes and concepts in the field RUL prediction, maintenance strategies, and related technologies (Fig. 2). An analysis using bibliometric mapping for review article enhances the quality of analysis by providing clear visual maps of keyword occurrences, temporal trend, cluster analysis and evolution in

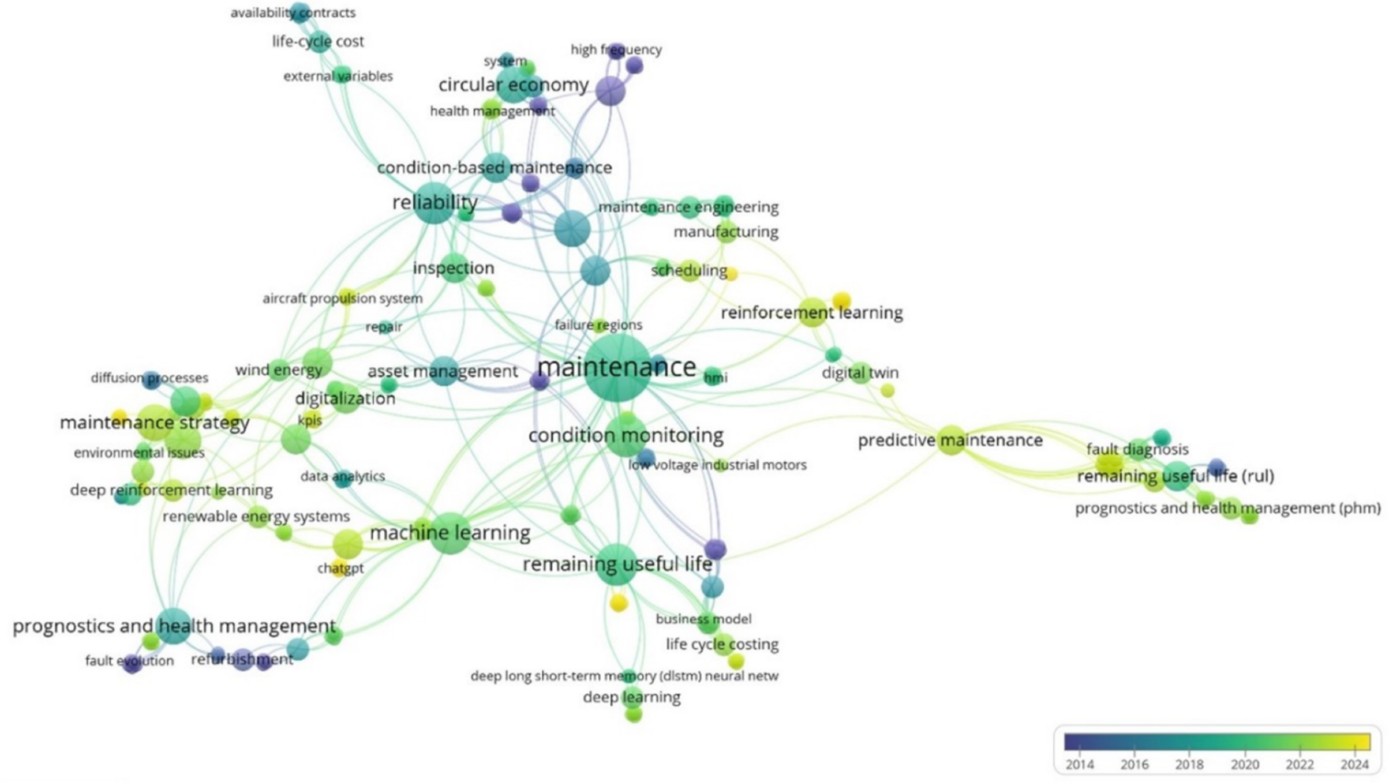

**Figure 2 Development of RUL key concepts since 2014.** Nodes: represent the key features of the map. Lines: lines between nodes represent the strength of relationships. Color: the development and prominence of concepts over time (2014–2024).

research interest (*Nobanee et al., 2021*). This allows authors to identify knowledge gaps, emerging themes, and research trajectories effectively. The frequency and quantity of publications associated with each concept are indicated by the size of the nodes, which represent key terms (*van Eck & Waltman, 2010*). The larger the node, the more publications are available online. The intensity of the relationships between nodes is reflected in the links between them, with closer distances indicating stronger connections (*Nees Jan & Waltman, 2014*). To emphasize the publication year and the intensity of their association, the links' thickness and colors are used.

The map has revealed significant trends over the past decade. The frequency of publications on RUL experienced a significant increase between 2019 and 2022, which is indicative of the increasing interest in predictive maintenance applications. From 2021, there was a significant increase in the study of machine learning, with a particular emphasis on its application in maintenance optimization. In the same vein, defect diagnosis gained prominence in 2020. Nevertheless, the map suggests that there is limited connectivity between these critical areas, despite the growth in these individual domains. This suggests an opportunity to further investigate their integration, such as the use of machine learning for fault diagnosis in RUL prediction. This emphasizes the necessity of

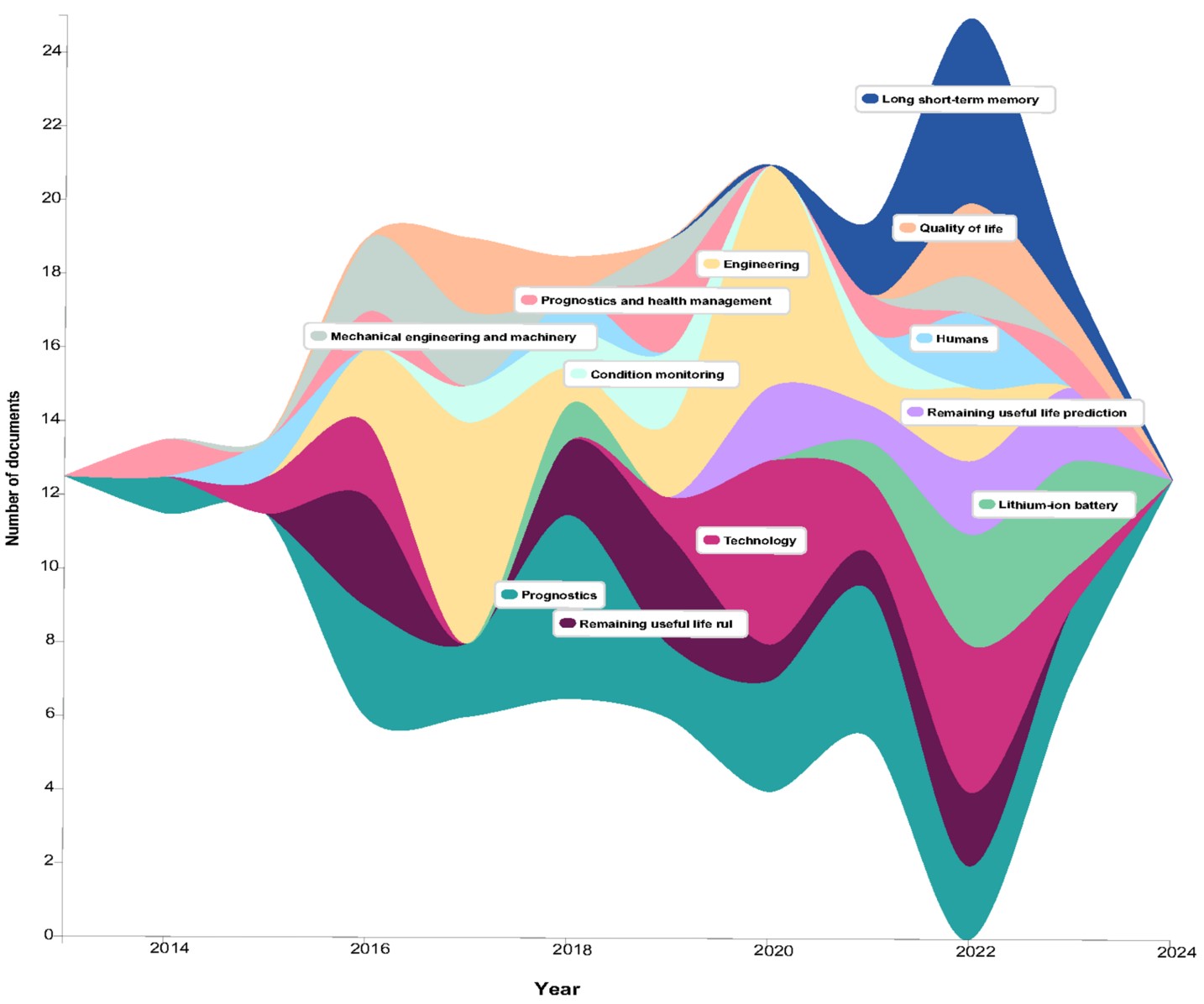

**Figure 3 Evolution numbers of RUL studies based on keywords over time.** X-axis: year of studies. Y-axis: number of documents.

conducting research that connects these categories in order to improve the comprehensive solutions for predictive maintenance.

This Streamgraph depicts the progression of research interests in RUL prediction and related disciplines over the past 10 years. It indicates a consistent and moderate level of interest between 2014 and 2018, with research covering a wide range of subjects, including mechanical engineering, condition monitoring, and prognostics (Fig. 3). However, a substantial decline was observed after 2020, which is likely attributable to the COVID-19 pandemic, which disrupted research activities on a global scale. Research interest in this particular field resumed in 2022, concurrently with the increasing emphasis on AI architectures, particularly LSTM

**Table 8 Distribution of RUL studies.**

| Life stage | Sub-stage | Failure rate | Failure | Article |
|---|---|---|---|---|
| **Early stage**<br>Infant mortality | Design manufacturing, licensing, establishment | High | Decrease over time as defective units are identified and repaired or replaced | *Zhang et al. (2023b)* |
| **Middle stage**<br>Normal life | Warranty period<br><br>Normal use<br><br>Heavy utilization<br><br>Upgrade/ modification | Low and constant | Randomly due to external factors | *Arunthavanathan et al. (2023)*, *Aydemir & Acar (2020)* to *Zheng, Liao & Zhu (2023)*, *Pei et al. (2019)* to *Wang & Zhao (2023)* |
| **Late stage**<br>Wear-out | Aging asset<br><br>Approaching end of life | High | Predictable, often due to aging process | *Arunthavanathan et al. (2023)*, *de Pater & Mitici (2023)*, *Zheng, Liao & Zhu (2023)*, *Pei et al. (2019)*, *Rosero, Silva & Ribeiro (2022)*, *Biondi, Sand & Harjunkoski (2017)*, *Ma, Xu & Yang (2023)* to *Solís-Martín, Galán-Páez & Borrego-Díaz (2023)* |

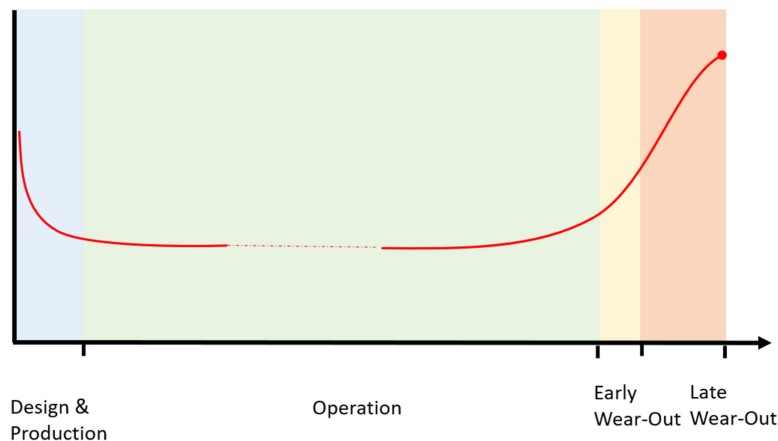

**Figure 4 The bathtub curve for equipment life cycle and failure rate.** X-axis: age of equipment; Y-axis: equipment failure rate over three phase of life; stage I, stage II and stage III.

networks. This surge is indicative of the growing incorporation of AI into RUL prediction, which demonstrates its capacity to improve prediction accuracy and offer valuable insights for maintenance strategies. The emergence of LSTM and its applications emphasizes the transition to data-driven approaches in the field, indicating a resurgence in research activity following the pandemic (*Open Knowledge Maps, 2024*).

## Synthesis and analysis based on research questions
### RQ1 What are the range of studies associated with predicting the RUL of equipment?
Table 8 measures studies interest distributed based on stage in life span where the bathtub curve model was used to identify the range of study because it defines components failure

rate over time (*Titu-Marius, 2021*). In Fig. 4, The bathtub curve typically comprises of three stages: an initial diminishing failure rate (infant mortality), a consistent failure rate (useful life), and a rising failure rate (wear-out) (*Ikonen, Corona & Harjunkoski, 2023*). Numerous models and methodologies have been devised to precisely elucidate and conform to this curve to empirical data. For example, the adjusted exponential-Weibull (MEW) model, which merges the exponential and Weibull distributions, presents a versatile approach to accommodating failure time data with a bathtub-shaped hazard rate, delivering superior outcomes in comparison to other models (*Al-Essa et al., 2023*).

The early stage is supported by one study *Zheng, Liao & Zhu (2023)*, indicating relatively less research focus on the high initial failure rates caused by manufacturing defects or early use issues. The absence of operational data is identified as a significant challenge, as RUL predictions heavily depend on this data, which is not available during the design and production stage since the equipment has not yet been put into use. In contrast, The operation stage has the most extensive research, with 19 studies. This extensive coverage indicates a well-established understanding of the low and normal constant failure rates during this phase, supporting effective maintenance strategies and reliability planning based on a broad consensus regarding random failures often due to external factors. The wear-out stage, which has been covered in eight studies, is mentioned divided into main two subphases: early wear out and late wear out stage. The transition phase marks the onset of increasing Major failure rates, accumulation of fatigue and outdated components. The late wear-out stage is defined by a sharp increase in failure frequency, signifies the approaching End of Life. Maintenance costs escalate, and decisions must be made regarding asset replacement, decommissioning, or life extension investments. This stage is particularly relevant for economic analysis comparing continued operation *vs.* replacement costs. Overall, the distribution of studies highlights well-researched middle and late stages, while the early stage presents an opportunity for further investigation to enhance early failure mitigation.

### RQ2 How can hybrid models combining physical and data-driven methods improve RUL prediction for complex systems?

Accurately predicting the RUL is crucial for ensuring equipment achieves its expected life span. Hybrid models, which integrate two or more methods, offer a promising approach to enhance prediction accuracy by combining both physical principles with data-centric techniques particularly in a complex system. A common hybrid approaches, integrates physical-based with data-driven model. Physical-based models rely on mathematical representations of failure mechanisms, providing a fundamental understanding of system behavior (*Wang & Zhao, 2023*). Combining support vector regression (SVR) and random forest regression (RFR) alongside an exponential weighted moving average (EWMA) control chart for anomaly detection (*Arunthavanathan et al., 2023*; *Wang & Mamo, 2019*).

Meanwhile data-driven models, utilized historical data to identify patterns and predict future states, offering flexibility and adaptability to varying conditions (*Wang & Zhao, 2023*). In time frequency data transformations, Short-time Fourier Transform (STFT) and Wigner Ville Distribution (WVD), used to analyze non-stationary signals, which are

common in complex systems (*Rosero, Silva & Ribeiro, 2022*). While in DL, CNN and RNN utilize in extracting multilevel features from data. PdM is a common example application of data-driven models. The integration with sensor and AI data processing technique enhanced the prediction and allow for faster decision making (*Sayyad et al., 2021*). Plus, a combine method will increase computational requirement and the accuracy of data-driven models hinges on the quality of their input data. Noisy data, often stemming from environmental influences, can significantly impact prediction accuracy (*Biondi, Sand & Harjunkoski, 2017*; *Fan, Nowaczyk & Rögnvaldsson, 2020*; *Ma, Xu & Yang, 2023*). Effective data pre-processing techniques are crucial to mitigate these issues and enhance model reliability. Furthermore, data-driven models must exhibit adaptability to varying operating conditions prevalent in industrial settings. Robust feature extraction methods and domain adaptation techniques are crucial to maintain prediction accuracy across diverse scenarios (*Wang & Zhao, 2023*).

### RQ3 What are the comparative accuracies between Artificial Intelligence approaches in RUL prediction?

Among the most effective machine learning methods for predicting equipment's RUL, deep learning techniques such as LSTM networks have demonstrated remarkable results (Table 9). LSTM combine with Autoencoders (LSTM-AE) produce high accuracy, provides monotonicity (0.38), trend ability (0.95), and prognosability (0.94). This highlights its ability to provide consistent health indicators outperforming other autoencoders such as gated recurrent unit autoencoder (GRU-AE) and bidirectional long short-term memory autoencoder (BiLSTM-AE) (*de Pater & Mitici, 2023*; *Wu et al., 2022*). On fault detection, the use of Robust-ResNet combined with LSTM and CNN architectures, achieves up to 99.53% accuracy (*Zheng, Liao & Zhu, 2023*). LSTM with COSMO features reduce mean absolute percentage error (MAPE) between 13–15%, compared to 25% with traditional methods (*Duan et al., 2023*). This approach capable in managing large amounts of temporal data and detecting subtle changes indicative of degradation or failure (*de Pater & Mitici, 2023*; *Fan, Nowaczyk & Rögnvaldsson, 2020*; *Wang & Zhao, 2023*; *Zheng, Liao & Zhu, 2023*).

In a situation of understanding data patterns in sequence, recursive neural network (RNN) is used. These in combination with the extended Kalman filter (EKF) reduces mean absolute error (MAE) by 15–25% and MAPE by 10–20% compared to standalone RNNs or traditional models, improving prediction accuracy for nonlinear, noisy datasets. While differentiating abnormality, OC-SVM used to distinguish abnormalities by separating them from normal operating conditions. The fault margin was dynamically adjusted from data patterns by considering the highest anomaly count in a window and incorporating a noise margin for accurate anomaly detection (*Arunthavanathan et al., 2023*).

CNN architecture was developed that incorporates domain adaptation techniques to minimize distribution discrepancies across different failure modes. Notable studies, such as those by *Cheng et al. (2021)*, *Rosero, Silva & Ribeiro (2022)*, *Solís-Martín, Galán-Páez & Borrego-Díaz (2023)* and *Ture et al. (2024)*, have highlighted the utility of CNNs in this domain. In TCNN, metrics assess higher accuracy and robustness RUL of bearings under

**Table 9  AI application and performance for health monitoring.**

| Method | Advantages | Applications | Prediction | Article | Significant result | |
|---|---|---|---|---|---|---|
| | | | | | Performances | Error measurement |
| Long short-term memory networks | Handle time-series data, capture temporal dependencies | Health indicators | RUL prediction | de Pater & Mitici (2023), | Accuracy: LSTM-AE 81–85% | RMSE 19% |
| | | | | Fan, Nowaczyk & Rögnvaldsson (2020) | Accuracy: PHM Score 0.85–0.92 | RMSE 15% to 25% MAPE <10% |
| | | | | Aydemir & Acar (2020) | Accuracy: LSTM (One Fault): 392 LSTM (Two Fault): 424 | RMSE (One Fault) 17.15 RMSE (Two Fault) 17.63 |
| | | | | Wang & Zhao (2023) | Accuracy: R-Square ($R^2$): 0.82 | Attention-GRU RMSE 24.90 MAE 16.4 cycles Attention-LSTM RMSE 22.40 $R^2$ 0.91 MAE 15.7 cycles |
| | | | | Sayyad et al. (2021) | Accuracy: LSTM model 92.54% | MAE 2.75 cycles RMSE 3.20 cycles |
| Convolutional neural network | Effective for processing visual data, extracting degradation features | Enhanced feature extraction | RUL prediction | Cheng et al. (2021), | Accuracy: R-square ($R^2$) 0.82 | MAE: 0.10 RMSE: 0.12 |
| | | | | Ture et al. (2024), | Accuracy: 93.93% | RMSE: 33.93 |
| | | | | Solís-Martín, Galán-Páez & Borrego-Díaz (2023) | NASA scoring function 0.015 Nil NASA scoring functions 2.13 | Bearing RMSE: 0.24 MAE: 0.17 Fast charging battery RMSE: 84.78 MAE: 51.98 Turbo engine RMSE: 10.46 MAE: 7.69 |
| | | | | Zheng, Liao & Zhu (2023) | Accuracy: Multi-channel: 81.74% Single-channel: 52.41% | – |
| | | | | Sayyad et al. (2021) | Accuracy: 89.56% | RMSE: 4.50 MAE: 3.10 |
| Stacking based ensemble learning | Enhances prediction accuracy by leveraging different base models | Improves prediction accuracy | RUL Prediction | Ture et al. (2024) | Accuracy: K-fold: 95.72% Leave one out: 95.69% | RMSE (K-fold): 33.25 RMSE (Leave one out): 31.30 |
| Dynamic Bayesian networks | Models' temporal processes with time-dependent variables | Predicting RUL of underwater self-enhanced structures with probability crack growth (PCG) | Anomaly detection | Liu et al. (2020) | Accuracy: 1st year: PCG 45.2% Crack value 0.4418 7th year: PCG 37% Crack value 4.7072 | 1st four years <8.5% 5th to10th years 10–20.4% After 10 years <11.3% |
| Wiener process models | Models' random phenomena with independent, normally distributed increments | Characterizes degradation trajectories, includes negative jumps | RUL prediction | Wang et al. (2018) | Accuracy: 80.51% | Model with Weiner process MAPE: 19.49% RMSE: 49.03 days MAE: 41.62 days |
| | | | | Zhang et al. (2020) | Accuracy: SSL estimation with spare part storage degradation: Final failure time: 170 | – |

| Method | Advantages | Applications | Prediction | Article | Significant result | |
|---|---|---|---|---|---|---|
| | | | | | Performances | Error measurement |
| Maximum likelihood estimation | Provides robust parameter estimates in nonlinear, non-Gaussian noise scenarios | Measures model's explanation of observed predictive maintenance data | RUL prediction | *Pei et al. (2019)* | Accuracy: Monitoring time scale 96.43% reduction then on natural time scale | MSE (Natural time scale): 22.99 MSE (Monitoring time scale): 0.82 |
| Least squares support vector machine | Constructs failure prediction models | Medical equipment failure prediction | Anomaly detection | *Ma, Xu & Yang (2023)* | – | AFS-ABC with SVM: error rate 2.5% in 0.85 s FMEA: error rate 5.3% in 1.23 s |
| Recursive neural network | Enhances prediction accuracy, reduces overfitting | Prediction tasks under complex conditions | RUL prediction | *Duan et al. (2023)* | – | MAE 11.83 MAPE 18.2% with Euclidean Distance: MAE 15.48 cycles MAPE 24.3% |
| Multi-layer Perceptron | Predicts failures based on historical data | Scheduling maintenance, reducing downtime+ | RUL prediction | *Rosero, Silva & Ribeiro (2022)* | Accuracy: Elbow Point (with HI): 72 h: 18% 48 h: 38% 36 h: 31% | RMSE (Without HI): 14.01 RMSE (with HI): 8.62 |
| Robust-ResNet | Fault detection, predicting RUL | Internal gear pumps analysis | RUL prediction | *Zheng, Liao & Zhu (2023)* | Accuracy: 99.53% | Error reduction: 17.79% |
| One-class support vector machine (OCSR) | Detects deviations from normal operating conditions | Identifying potential faults | Anomaly detection | *Arunthavanathan et al. (2023)* | Accuracy: Ordinary least squares: 98.55% Bayesian linear regression: 99.51% | Reactor cooling tower result: OLSR predicted RUL: 12,345 samples (compared to the actual 18,909 samples) Condenser cooling water valve stiction result: OLSR predicted RUL: 7,123 samples (compared to the actual 9,451 samples). |

multiple failure behaviors with a low MAE of 0.10, root mean square error (RMSE) of 0.12, and a high R-squared ($R^2$) of 0.82 (*Cheng et al., 2021*). CNN has achieved an accuracy of 93.93% performed well among DL models but slightly lower than stacking based ensemble learning (95.72%) (*Ture et al., 2024*).

Dynamic Bayesian networks (DBN), and Wiener process models offer robust frameworks for modeling temporal and degradation processes in corrective maintenance. DBN incorporates time-dependent variables and their probabilistic dependencies, enabling effective representation of dynamic systems and their evolution over time. However, Wiener process models characterize degradation trajectories by accounting for both gradual wear and negative jumps caused by imperfect maintenance. These approaches as discussed in studies *Wang et al. (2018)*, *Liu et al. (2020)* and *Zhang et al. (2020)* provide robust tools for understanding and predicting equipment performance

under varying operational conditions. Through a comparison with experimental data, DBN shows an error margin lower than 8.5% in the first 4 years, lower than 20.4% between 5–10 years and lower than 11.3% after 10 years (*Liu et al., 2020*).

Moreover, the use of Kalman filtering not only made an average improvement 18% in prediction results, but also reduce 10–15% compared to static methods like maximum likelihood estimation (MLE) (*Pei et al., 2019*). The least squares support vector machine (LS-SVM) algorithm with the artificial fish swarm—artificial bee colony algorithm (AFS-ABC) increase prediction in potential failures and assess equipment health. This model has accomplished lower error forecast compared to other methods (*Ma, Xu & Yang, 2023*). Bayesian methods, which involve developing exponential degradation models and updating parameters using real-time condition monitoring data, offer another effective approach (*Liao & Tian, 2012*).

Bar chart review methods in grouped categories based on functions (Fig. 5). For time-series analysis, methods like LSTM and RNN help track changes over time (four methods). Feature extraction, like CNN and ResNet, pulls useful patterns from data (two methods). Statistical modeling, such as Kalman filtering and Wiener process, uses mathematical approaches to predict outcomes (two methods). Anomaly detection, like OC-SVM and LS-SVM, identifies unusual patterns (two methods). Ensemble methods, like stacking, combine multiple models for better accuracy (one method) and general models, such as MLP and ResNet, are flexible tools for general predictions (two methods). These visualizations offer a well-rounded understanding of how different machine learning techniques, with components ranging from 1 to 4, utilize varying levels of complexity in their approaches to RUL prediction.

The high dependency on AI algorithms raises the risk of algorithmic biases, impact the fairness and reliability of these systems. Despite their prevalence, none of this research mentioned on the influence of these biases or proposed mitigation strategies. Algorithmic models may also reinforce historical inequalities by assuming that past trends predict future outcomes, as seen in recommender systems that limit diversity by continuously suggesting similar products.

### RQ4 How effective are FDD approaches in detecting early indicators of equipment failure?

Figure 6 shows an overall schematic flow of FDD integration to anomaly detection. The integration of advanced technique such as FDD has significantly enhanced the ability to detect early indicators of equipment failure (*Rosero, Silva & Ribeiro, 2022*). Over the years, the technique has evolved and categorize into mathematical, analytical, data-driven, statistical, computational, and hybrid approaches (*Zhao et al., 2019*). To establish into one of these methods, variables associated with the fault need to be identified in advance. Abnormal values are detected by manually setting a threshold for these variables. These thresholds vary across different system components and must be adjusted for each specific application (*Arunthavanathan et al., 2023*; *Nelson & Culp, 2023*).

Table 10 encapsulates classification for fault detection studies in FDD strategy. In data-driven methods *Carroll et al. (2019)* and *Wang & Mamo (2019)* demonstrates the
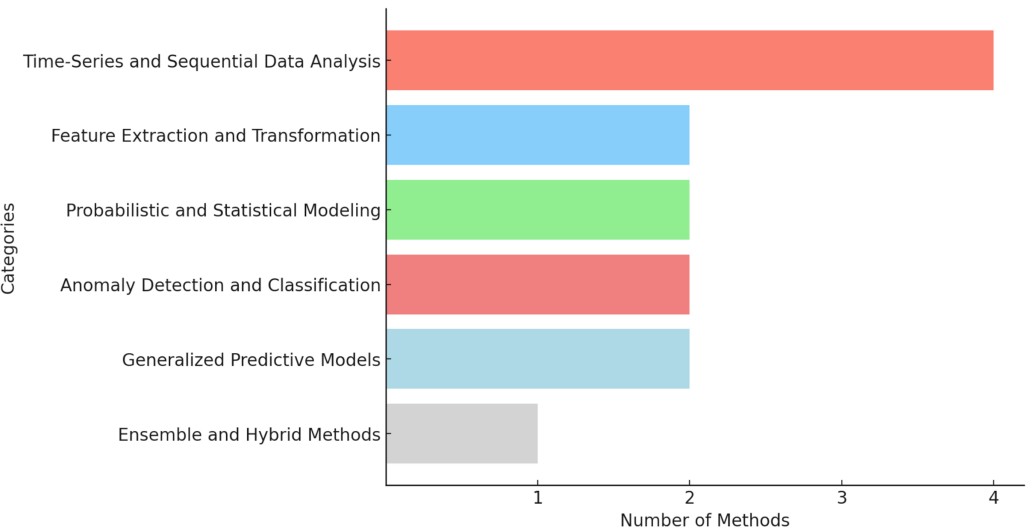

**Figure 5  Number of components involve in AI.** Grouping of methods by functional categories. X-axis: number of methods. Y-axis: categories.               

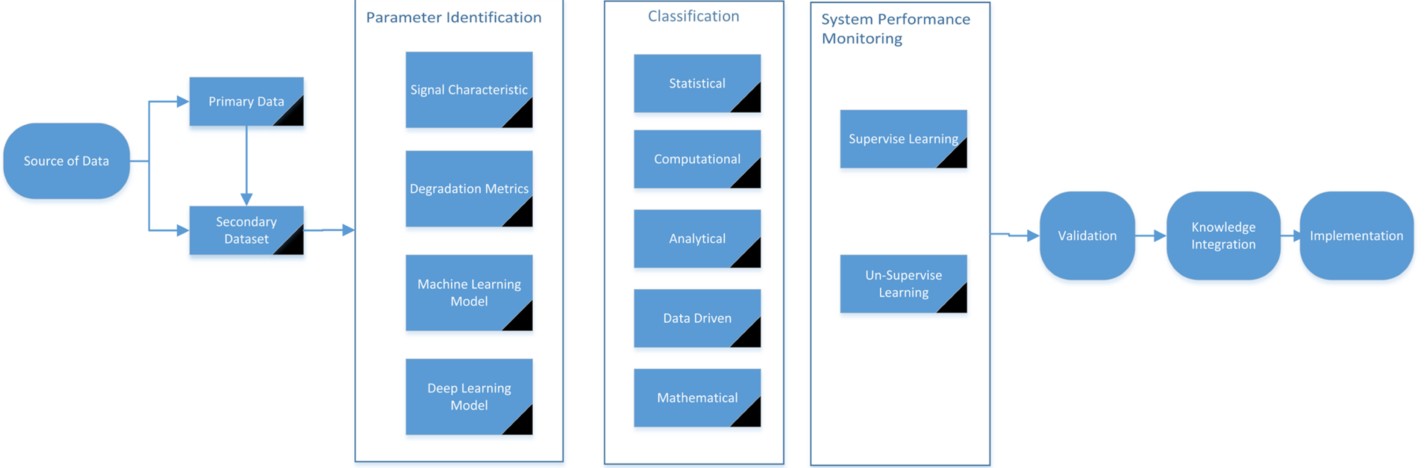

**Figure 6  FDD integration with anomaly detection flowchart.**               

**Table 10  Method in FDD.**

| Classification | Total | Article |
|---|---|---|
| Statistical and mathematical method | 13 | *Arunthavanathan et al. (2023)*, *Aydemir & Acar (2020)*, *Wang et al. (2018)*, *Zhang et al. (2020)*, *Pei et al. (2019)*, *Carroll et al. (2019)*, *Biondi, Sand & Harjunkoski (2017)*, *Wang & Mamo (2019)* to *Solís-Martín, Galán-Páez & Borrego-Díaz (2023)* and *Liao & Tian (2012)* |
| Data-driven and analytical method | 19 | *Arunthavanathan et al. (2023)*, *Aydemir & Acar (2020)* to *Solís-Martín, Galán-Páez & Borrego-Díaz (2023)* and *Liao & Tian (2012)* |
| Model based method | 18 | *Arunthavanathan et al. (2023)*, *Aydemir & Acar (2020)* to *Zheng, Liao & Zhu (2023)*, *Pei et al. (2019)* to *Ture et al. (2024)*, *Carroll et al. (2019)* to *Sayyad et al. (2021)*, *Lee, Kim & Lee (2023)* to *Solís-Martín, Galán-Páez & Borrego-Díaz (2023)*, *Wang & Zhao (2023)* and *Liao & Tian (2012)* |

real-time sensor to reflect the dynamic behavior of systems efficiency, combine with artificial neural networks (ANNs) in predicting gearbox failures using SCADA and vibration data to achieved higher accuracy particularly when using high-frequency vibration data (*Carroll et al., 2019*). *de Pater & Mitici (2023)* focus on LSTM autoencoders, showing that the reconstruction error, which increases with system degradation, can effectively identify early faults. Statistical methods are highlighted by *Aydemir & Acar (2020)* discuss the use of the CUSUM control chart for anomaly detection, which helps identify the degradation onset point and subsequently improves the performance of RUL estimators by focusing on active degradation periods. *Wang & Mamo (2019)* employ the EWMA control chart for anomaly detection, which is crucial in their hybrid approach, where detected anomalies trigger RUL prediction models, thereby enhancing early fault detection. Model-based methods such as those described by *Ma, Xu & Yang (2023)* utilize deterministic reasoning and fuzzy inference for precise and uncertain data respectively, achieving effective fault detection. Hybrid methods, including the approaches by *Arunthavanathan et al. (2023)*, *Lee, Kim & Lee (2023)*, combine techniques like OC-SVM for fault detection with neural network permutation algorithms for classification, and use alarm rules alongside contextual diagnosis to enhance fault detection accuracy.

System performance monitoring further model known faults (as supervised learning), while unsupervised learning detects to unknown anomalies or unusual patterns in the data. This real-time monitoring ensures that even early signs of failure are noticed. Finally, the models are tested during validation to ensure they work accurately, and the insights are stored in a knowledge base for future use. Once validated, the system is deployed in the implementation stage, where it operates as part of real-time monitoring to predict and prevent equipment failure effectively. Unlike existing reviews, this assesses the FDD efficiency, systematically evaluates their performance across various contexts, accentuating pragmatic applications. Whereas the most of the reviews concentrate on singular methodologies or particular domains, highlighting the integration of hybrid and data-driven techniques, which are often overlooked in similar studies, the review underscores the amalgamation of data-centric and hybrid strategies, offering an all-encompassing view of their real-world ramifications.

### RQ5 What challenges are associated with the implementation of sensor technology for fault detection?

One of the methods in early detection equipment malfunction is implementing sensors as a primary data in FDD. This proactive approach minimizes downtime, reduces costs, and improves safety across various applications. Table 11 summarizes the integration of multiple sensors for continuous monitoring.

*Deterioration measurement*

At the stage period, the focus is essential on capturing wear and tear through deterioration measurements (*Sayyad et al., 2021*; *Solís-Martín, Galán-Páez & Borrego-Díaz, 2023*). Accelerated degradation testing (ADT) is often employed, subjecting equipment to harsher-than-normal operating conditions to model degradation behaviors

**Table 11 Sensor integration of data driven method for data collection.**

| Sensor parameter | Article | Measurement | Benefits | Limitations |
|---|---|---|---|---|
| Deterioration | *Liao & Tian (2012)*, *Zhang et al. (2020)*, *Sayyad et al. (2021)*, *Solís-Martín, Galán-Páez & Borrego-Díaz (2023)*, *Fan, Nowaczyk & Rögnvaldsson (2020)*, *Wang & Zhao (2023)* | Wear and tear | Predictive maintenance, monitoring | Performance effect and increasing noise factor |
| Vibration | *Aydemir & Acar (2020)*, *Zheng, Liao & Zhu (2023)*, *Cheng et al. (2021)*, *Wang & Mamo (2019)*, *Lee, Kim & Lee (2023)* | Detect imbalance and alignment | Effective measures changes in mechanical condition, can detect wide range of faulty condition | Require proper mounting to ensure accurate reading and sensitive to environmental noise |
| Temperature | *Aydemir & Acar (2020)*, *Cheng et al. (2021)*, *Rosero, Silva & Ribeiro (2022)*, *Carroll et al. (2019)* | Overheating level and thermal stress | Cost effective, provide early warnings of thermal issue | Limited to surface temperature measurement, can't capture internal component temperature accurately |
| Corrosion | *Liu et al. (2020)*, *Ture et al. (2024)* | Integration physical models alongside stress mechanics to define rate of metal loss | Prevent catastrophic failure and reduce safety hazards | Environmental factor could lead to accelerated noise |
| Pressure | *Aydemir & Acar (2020)*, *Zheng, Liao & Zhu (2023)*, *Ture et al. (2024)* | Hydraulic pressure level Weight distribution | Essential for monitoring fluid system | Requires calibration, result effected by environmental condition |
| Abnormality trigger | *de Pater & Mitici (2023)* | Health indicator | Early detection of unhealthy stage | Generalization leads to unknown types of failure |
| Acoustic | *Solís-Martín, Galán-Páez & Borrego-Díaz (2023)* | Soundwave from abnormalities | Less sensitive to environmental, easy to deploy | Sensitive to environmental noise, need signal processing |
| Operating time | *Pei et al. (2019)* | Nonlinear degradation | Equipment schedule maintenance, asset utilization | Time consuming |
| Crack | *Liu et al. (2020)* | Stress distribution and fatigue growth | Prevent catastrophic failure and reduce safety hazards | Limited detection sensitivity and require combination with other measurement |

(*Liao & Tian, 2012*). Additionally, emphasis is placed on linking degradation during operation, storage, and the recovery state after replacement (*Zhang et al., 2020*). Advanced techniques such LSTM networks and CNN are utilized to process from sensor data and identify patterns of degradation. This beneficial and performance can be significantly impacted accuracy of the predictive models (*Fan, Nowaczyk & Rögnvaldsson, 2020*; *Wang & Zhao, 2023*).

*Vibration monitoring*
Vibration sensors are highly effective in diagnosing health machine, by detecting vibrations caused by friction, tool wear, or fractured inserts between the tool and workpiece during operation (*Aydemir & Acar, 2020*; *Sayyad et al., 2021*). High-frequency accelerometers are used to measure vibration signals, capturing horizontal and vertical movements (*Cheng et al., 2021*). Metrics such as vibration amplitude, frequency

components, and acceleration are extracted to create indicators (*Zheng, Liao & Zhu, 2023*). Vibration data serves to detect imbalance and misalignment in mechanical systems (*Lee, Kim & Lee, 2023*; *Wang & Mamo, 2019*). This method is highly effective in capturing changes in mechanical conditions and diagnosing a range of faulty conditions. However, proper sensor mounting is crucial to avoid errors, and the sensors are often susceptible to environmental noise, requiring additional filtering and processing for accurate readings.

*Temperature sensors*
Temperature sensors can detect overheating and thermal stress, providing cost-effective and early warnings of potential thermal issues. ML approaches, particularly deep learning algorithms like LSTM networks, been utilized for time-based inputs and predict RUL by learning from sensor data, including temperature measurements (*Aydemir & Acar, 2020*). As cited in articles (*Carroll et al., 2019*; *Cheng et al., 2021*; *Rosero, Silva & Ribeiro, 2022*) temperature monitoring is employed to assess overheating levels and thermal stress in equipment. This cost-effective method provides early warnings of thermal issues. However, it is limited to surface temperature measurements and may fail to capture internal temperature variations, leading to incomplete diagnostics.

*Corrosion assessment*
Corrosion data integrates physical models alongside stress mechanics to determine the rate of metal loss. Corrosion together with vibration sensor collect primary data, such as stress related degradation or material thickness reduction due to rust (*Ture et al., 2024*). This measurement helps prevent catastrophic failures and enhances safety by reducing hazards (*Liu et al., 2020*). However, external environmental factors can introduce noise, accelerating the degradation rate and complicating accurate assessments.

*Pressure monitoring*
*Aydemir & Acar (2020)*, *Zheng, Liao & Zhu (2023)* and *Ture et al. (2024)* highlight the importance of pressure data for evaluating hydraulic pressure levels and weight distribution in fluid systems. Critical parameters such as outlet pressure, internal system pressure, and operational load in machinery are used to monitor. CUSUM is a statistical tool employed with pressure sensor data to detect significant deviations, marking the onset of degradation. It is highly effective in identifying gradual changes that might not be apparent in raw sensor data (*Aydemir & Acar, 2020*). Pressure sensors measure pulsation signals, directly reflect non-uniform fluctuations within the pump and are essential for early fault detection (*Zheng, Liao & Zhu, 2023*). This parameter is crucial for maintaining fluid system health but requires regular calibration. Furthermore, environmental conditions can significantly affect the accuracy of pressure readings.

*Abnormality triggers*
As discussed in *de Pater & Mitici (2023)* anomaly triggers (AT-AE) planted for detecting the early stages of equipment failure. They are effective in signaling unhealthy operational

conditions but may generalize across fault types, leading to unidentified or unaccounted-for failures.

*Acoustic sensors*

Pressure and acoustic sensors can detect abnormal sound waves that indicate potential faults, offering a less environmentally sensitive option that is easy to deploy. For instance, feature extraction from hydraulic pressure signals using techniques like complementary ensemble empirical mode decomposition (CEEMD) and singular value decomposition (SVD) has been employed to construct feature vectors for fault diagnosis in hydraulic pumps (*Nelson & Culp, 2023*). Examines the role of soundwave detection in identifying abnormalities. Acoustic sensors are less affected by environmental conditions, but they are highly sensitive to external noise, requiring sophisticated signal processing for reliable interpretation.

*Operating time*

Operating time data monitors non-linear degradation trends. It is useful for scheduling maintenance and optimizing asset utilization (*Pei et al., 2019*). However, the process is time-intensive and requires significant computational resources.

*Crack detection*

Stress distribution and fatigue growth are monitored to prevent catastrophic failures. Crack detection enhances safety and reliability but has limited sensitivity and often requires integration with other measurement methods for comprehensive diagnostics (*Liu et al., 2020*).

Moreover, humidity sensors are critical for monitoring moisture levels, preventing corrosion, and maintaining the integrity of materials, though they require regular calibration and are sensitive to dust. Optical sensors, which measure light intensity, are effective for detecting changes in lighting conditions and are non-intrusive, though they require regular cleaning to maintain accuracy (*Solís-Martín, Galán-Páez & Borrego-Díaz, 2023*).

### RQ6 What are the most effective methods use for monitoring condition in the predictive maintenance and how can root cause analysis be effectively determined in fault tolerance systems?

Table 12 highlight method currently practice in monitoring asset and system. One of the most common effective methods is predictive maintenance (PdM), which utilizes sensor data and advanced analytics to predict equipment failures before they occur. *Wang et al. (2018)* emphasized the importance of predictive maintenance of degrading systems, thereby improving overall reliability and maintenance scheduling. *Aydemir & Acar (2020)* demonstrated that anomaly monitoring significantly improves RUL predictions, ensuring timely and effective maintenance interventions. Condition-based monitoring (CBM) is another effective method that involves continuous or periodic monitoring of equipment condition using sensors to detect deviations from normal operation. *Zheng, Liao & Zhu (2023)* developed a fault detection model for internal gear pumps, which enhances the

**Table 12 Maintenance management and fault tolerance monitoring method.**

| Method | Article | Focus area | Application in monitoring |
|---|---|---|---|
| Machine learning techniques | Cheng et al. (2021) | Integration with statistical degradation models | Predictive maintenance, risk management |
| Robust health indicators | Rosero, Silva & Ribeiro (2022) | Development of robust health indicators that can predict RUL accurately under varying conditions and limited failure data. | Maintenance planning and decision |
| Enhanced RUL predictions | Aydemir & Acar (2020) | Triggering estimation post-degradation detection | Maintenance planning, system reliability |
| Imperfect maintenance consideration | Wang & Zhao (2023) | Accounting for imperfect maintenance | Maintenance planning and decisions |
| Anomaly detection integration | Aydemir & Acar (2020) | Combining anomaly detection with machine learning | Preventing unexpected failures |
| Pressure self-enhancement effects | Fan, Nowaczyk & Rögnvaldsson (2020) | Study of pressure effects | Maintenance planning and decision, system reliability |
| Proactive maintenance strategies | Aydemir & Acar (2020) | Supported by accurate RUL predictions | System reliability |
| Feature selection process | Duan et al. (2023) | Improving feature selection for RUL prediction | Maintenance planning and decision |
| Transfer learning and domain adaptation | Solís-Martín, Galán-Páez & Borrego-Díaz (2023) | Adapting to varying conditions | Maintenance planning and decisions |
| Bayesian approach for real-time applications | Ma, Xu & Yang (2023) | Continuous prediction updates | Real-time applications |

effectiveness of CBM by accurately detecting faults and predicting RUL. These advanced monitoring techniques, combined with the use of machine learning and AI, such as the work by Cheng et al. (2021) using transferable convolutional neural networks, provide robust solutions for fault detection and RUL predictions. Root cause analysis (RCA) is a systematic method that involves collecting data, analyzing failure modes, and identifying the underlying reasons for faults. This approach ensures that the real cause of the problem is addressed rather than just treating the symptoms. Arunthavanathan et al. (2023) highlighted the significance of RCA in estimating RUL and transforming fault-to-failure processes in process systems. Fault tree analysis (FTA) and failure mode and effects analysis (FMEA) are additional methods that support RCA by providing structured frameworks for identifying and prioritizing potential causes of failures.

### RQ7 How do life cycle analysis and survival analysis frameworks influence the selection of maintenance strategies and health indices in reliability driven maintenance?

Life cycle analysis (LCA) and survival analysis frameworks significantly influence the selection in reliability-driven maintenance by providing a structured proactive and reactive approach to evaluate and optimize maintenance decisions. Preventive maintenance (PM) is characterized by its proactive approach, involving regular inspections, servicing, and timely interventions to prevent equipment failures. While corrective maintenance (CM) is reactive approach, initiated only after equipment has failed. This approach solves the issues in the short term but leads to higher overall costs and reduced equipment lifespan.

According to *Wang et al. (2018)*, continuous PM can substantially extend the lifespan of equipment, prolong RUL and predict failures before it occurs (*Pei et al., 2019*). Effective maintenance scheduling and consequently, a longer equipment lifespan (*Aydemir & Acar, 2020*). CM is reactive, initiated only after equipment has failed. While this strategy might seem cost-effective in the short term due to lower initial maintenance expenditures, leads to higher overall costs and reduced equipment lifespan. *de Pater & Mitici (2023)* assert that PM's focus on timely and planned interventions not only improves reliability but also optimizes maintenance resources. *Cheng et al. (2021)* illustrate how integrating AI with PM protocols enhances RUL predictions and fault detection accuracy.

CM results in extended downtimes and higher repair costs because failures are addressed only after they have occurred, often leading to significant damage (*Sayyad et al., 2021*). *Carroll et al. (2019)* mentioned the unpredictable nature of failures under CM necessitates expensive emergency repairs and replacements, further diminishing the equipment's operational life. *Ture et al. (2024)*, demonstrated that the implementation of predictive maintenance algorithms within PM frameworks significantly reduces unexpected failures and maintenance costs, thereby extending the operational life of assets.

In the context of equipment reliability, life cycle analysis helps identify potential failure points and maintenance needs at different stages of the equipment's life. *de Pater & Mitici (2023)* demonstrated by understanding the degradation effect of wear and tear item, the implementation of predictive maintenance strategies will help preventing unexpected failures and enhancing overall reliability. Integrating LCA with cost analysis allows for the identification of the most cost-effective maintenance interventions. *Aydemir & Acar (2020)* use CUSUM in anomaly detection techniques, an integration LCA with cost analysis to enhances the accuracy of RUL estimation.

In survival analysis, reliability measured in statistical approach, focusing on predicting the time until a system fails based on its current condition and operational history (*Arunthavanathan et al., 2023*). The analytical approach places emphasis on survival function analysis, time to failure analysis and hazard function analysis.

*Time to failure analysis*

According to *Wang et al. (2018)*, analysis models can incorporate various factors influencing equipment degradation. By utilizing historical failure data, survival analysis helps in forecasting future failures, enabling proactive maintenance actions. *Liu et al. (2020)* through the analysis of degradation patterns, it becomes feasible to accurately forecast the RUL. *Cheng et al. (2021)* indicate that the utilization of TCNN can adjust to diverse failure patterns, rendering the models highly efficient across various machinery types. *de Pater & Mitici (2023)* demonstrate that LSTM autoencoders, capable of learning from limited failure data and adjusting to diverse circumstances, offer dependable RUL predictions for systems with scarce failure records.

*Survival function analysis*

The process begins with collecting data from various sensors, including vibration and pressure pulsation signals, during the initial performance tests and throughout the pump's

operational life (*Zheng, Liao & Zhu, 2023*). These advanced models help in identifying subtle signs of wear and tear that might be overlooked by traditional methods, thereby enhancing the reliability of equipment.

### Hazard function analysis

*Aydemir & Acar (2020)* emphasizes the advantages of anomaly detection in enhancing RUL estimation. By identifying deviations from normal operations at an early stage, maintenance activities can be strategically scheduled, averting breakdowns and guaranteeing equipment dependability. However, in this analysis, identifying defect in time-varying conditions and nonlinear conditions is not sufficient due to its complexity. Therefore, the needs for secondary and tertiary analysis needed before any decision is made. Health indices can act as a critical bridge between life cycle analysis and survival analysis in maintenance decision-making processes.

### Condition monitoring

These indices provide actionable insights for implementing condition-based maintenance strategies. The health indicators are evaluated using metrics such as monotonicity, trend-ability and prognostic-ability, measure the consistency, correlation with time and consistency across different units, respectively, providing clear signals of system degradation (*de Pater & Mitici, 2023*). One common approach is using root mean square (RMS) values of vibrations as health indicators, as demonstrated in the study where the RMS of horizontal vibration was selected for further analysis due to its significant correlation with the health state of bearings (*Wang & Mamo, 2019*). Another method involves PCA to simplify computations while retaining maximum original information. PCA standardizes sensor data, calculates covariance matrices, and projects data onto principal components to derive preliminary health indicators (*Duan et al., 2023*). LSTM autoencoders, employed to learn normal system behavior from unlabeled data and detect deviations indicative of degradation. Reconstruction errors from these models serve as health indicators, with variations including linear regression and Gaussian distribution models (*de Pater & Mitici, 2023*). Empirical mode decomposition (EMD) is another technique where the first intrinsic mode function (IMF) derived from time series data represents the HI, capturing the evolution of health conditions over time (*Rosero, Silva & Ribeiro, 2022*). Deep convolutional neural networks and recurrent neural networks, are used to map sensor inputs to HIs, which are then mapped to RUL. Techniques like stochastic modeling and distance-based approaches also contribute to HI calculation, with some methods simulating degradation paths using PCA space or employing exponential models for data-level fusion (*Fan, Nowaczyk & Rögnvaldsson, 2020*).

### Real time reliability metrics

In maintenance management, particularly those incorporating PHM, continuous tracking and evaluate equipment status through data collection, real-time monitoring, and fault diagnosis, which allows for early detection of potential risks and effective maintenance planning (*Duan et al., 2023*). The system itself designed to collect and analyze data from various sensors and monitoring devices installed on equipment. Parts of its objectives to

estimate equipment failure, reduce downtime thus allows for more effective maintenance planning and lower maintenance costs significantly. These systems utilize PHM to detect anomalies that trigger RUL estimation (*Aydemir & Acar, 2020*). By continuously monitoring, Systems can identify early signs of wear and tear, degradation effect and allow for maintenance teams to address issues early before major failures happen (*Ture et al., 2024*).

*Failure threshold*

Incorporating insights from both frameworks. PdM strategies, which are a subset of PHM, use RUL concepts to estimate the remaining time an equipment can function without failing, thus preventing unexpected downtimes and reducing maintenance costs (*Rosero, Silva & Ribeiro, 2022*; *Solís-Martín, Galán-Páez & Borrego-Díaz, 2023*). Integrating health indices into maintenance management systems enhances the safety and reliability of operations by providing continuous oversight and timely alerts for maintenance needs (*Ture et al., 2024*). Overall, maintenance management systems and health indices are integral to reliability-driven maintenance as they enable proactive maintenance strategies, extend the lifespan of components, and ensure the smooth operation of systems by providing timely and accurate predictions of equipment health and performance (*Duan et al., 2023*; *Ture et al., 2024*; *Wang & Zhao, 2023*).

# DISCUSSION

RUL prediction is a crucial aspect in preventing equipment malfunctions and reducing maintenance costs, with AI algorithms being a popular choice due to their flexibility and convenience (*Heng et al., 2009*; *Lei, 2016*). However, these algorithms often require large datasets and feature selection of hyper-parameters for optimal performance. Classical machine learning is not adequate to learn from these data, a task that presents unique challenges (*Calabrese et al., 2022*; *He et al., 2023*). The prediction techniques for estimating the lifespan of equipment vary significantly depending on the stage and equipment's condition; methods use for newly developed equipment may differ from those applied to equipment in active use or aging equipment (*Zhang et al., 2023b*).

## Research prospect

### Predicting RUL multiple asset under different stage of life span

Predicting the life span in the early stages of a component's life is crucial for newly develop product, this to preventing unexpected failures in the early design stages (*Haobin, Zhang & Sinha, 2024*). Table 13 display predictable test used in different stages of the equipment's life (*Qian, Yan & Gao, 2017*; *Kim et al., 2004*). Accelerated life testing (ALT) is a critical methodology used in manufacturing with its primary goal to accelerate the aging process of components, thereby obtaining significant life span data in a shorter period (*Qiu & Li, 2024*). In the middle stage, multiple factors involve in life prediction from utilization of usage, corrective maintenance, replacement parts and upgrading is a multifaceted challenge that requires integrating various advanced methodologies (*Noot, Martin & Birmele, 2025*). A multi-stage maintenance-impact degradation model based on the

**Table 13 Equipment prediction test under different stage of life span.**

| Objective | Stage of equipment | Prediction test | Dataset required |
|---|---|---|---|
| Life span prediction | Early stage | Accelerated life test Simulation test | Environment factor |
| | | | Component test data |
| | | | Quality test data |
| | Middle stage | Operational and utilization | Utilization data |
| | | Maintenance | Corrective maintenance data |
| | | Downtime action | Historical part replacement |
| | | Upgrading | Hardware and software upgrade |
| | Late stage | End of life | Aging data |
| | | | Salvage value |
| | | | Disposal data |
| Remaining useful life | Equipment/Component | Degradation method | Wear and tear data |
| | | | Life span prediction (stage of equipment) |

Wiener process can account for dynamic maintenance and failure thresholds, thereby improving the precision of RUL predictions (*Li et al., 2024*). Addressing uncertainties in RUL estimation is essential for improving the reliability and accuracy of predictive methods. These uncertainties fall into two main categories: epistemic and aleatory. Epistemic uncertainty comes from limited knowledge or incomplete information about the system, while aleatory uncertainty is due to the inherent randomness and variability in the system's behavior. Various strategies have been proposed to tackle these uncertainties (*Cao & Peng, 2023*). Model-based strategies frequently encounter difficulties with complicated connections and uncertainties, whereas data-driven approaches sometimes neglect previous knowledge and struggle with restricted data (*Liang, Liu & Xiao, 2024*).

Integration of FDD and RUL estimation for the early detection of system faults and the prediction of the system's future operational life, facilitating timely maintenance actions and reducing unexpected downtimes. A novel tree network framework can address fault classification and RUL prediction in parallel, improving model selection accuracy and prediction efficiency (*Chai et al., 2024*). Similarly, a joint learning model that simultaneously performs failure mode recognition and RUL prediction by leveraging multiple sensor signals has shown promising results (*Wang, Xian & Song, 2023*). Combining system modeling methods with regression-based approaches and genetic programming algorithms to predict fault occurrences and estimate RUL, even in the presence of measurement noise (*Bahareh & Jørn, 2023*). A range of studies have explored the use of probabilistic, highlight the importance of considering uncertainties in input variables. *Zamzam et al. (2021)* established a ranking assessment, prioritizing and predictive systems both for medical equipment maintenance, using machine learning algorithms for medical equipment. These studies collectively underscore the potential of probabilistic models in improving the accuracy and effectiveness of life cycle cost analysis in the medical equipment domain.

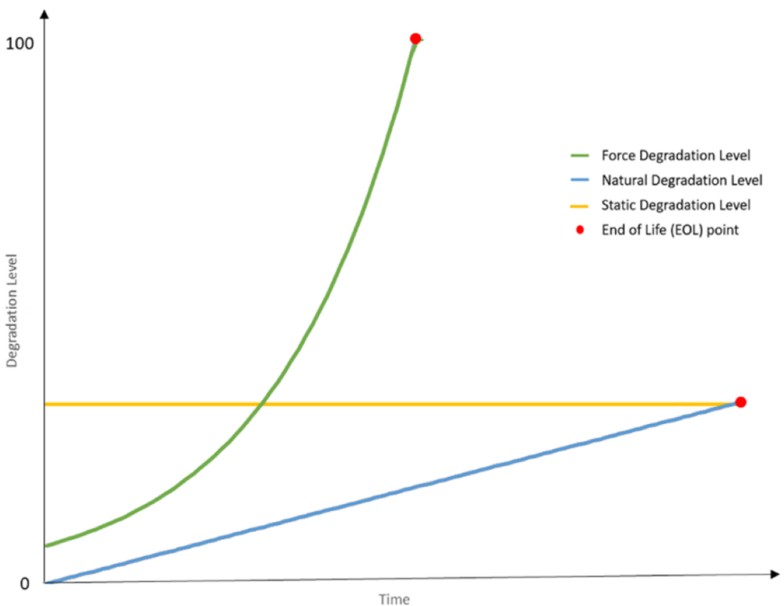

**Figure 7 State of degradation on equipment physical surface.** X-axis (Time): progression of time. Y-axis (Degradation level): the degree of degradation. Line chart: green line: force (accelerated) degradation level. Blue line: natural degradation level. Yellow line: static degradation level red dots (EOL point): mark the point where equipment can no longer maintain its intended function due to excessive degradation.

### Uncertainty in equipment deterioration

There is often an incomplete information about all the factors influencing equipment deterioration. Models used to predict deterioration often rely on assumptions and historical data, which may not account for all possible future scenarios, leading to uncertainty in predictions (*Zhu et al., 2024*). State of degradation on equipment may consist of multiple types, including natural degradation, static degradation, and force-induced degradation, each contributing differently to the equipment's end-of-life (Fig. 7). Natural degradation refers to the gradual wear and tear that occurs over time due to regular usage and environmental factors (*Li et al., 2023*).

Performance degradation assessment (PDA) methods, which utilize statistical and intrinsic energy features, are crucial for evaluating such static degradation states by constructing high-dimensional feature sets and reducing them to sensitive health indices (*Lv, Hu & Wang, 2023*). *Zhang et al. (2020)* emphasize the role of health indicators in maintenance strategies by enabling more effective scheduling of maintenance activities. *Zheng, Liao & Zhu (2023)* develops health indicators based on the fault types classification and RUL stages, thus allowing for timely intervention. Further models and methods enhancement have been developed to enhance the predictions to support maintenance planning. For example, *Pei et al. (2019)* and *Duan et al. (2023)* highlight the importance of accurate RUL estimation for maintenance management, remarking that reliable predictions enable better scheduling of maintenance activities and improve the overall reliability of systems. The integration of ML and Internet of Things (IoT) has transformed

maintenance management systems into a comprehensive solution, allowing for real-time data processing and more sophisticated analysis of health indices (*Cheng et al., 2021*; *Lee, Kim & Lee, 2023*).

### Digital twin and virtual assistance

The integration of digital twin (DT) and virtual assistance (VA) technologies ML is revolutionizing fault identification, monitoring prediction in asset management. This is crucial in assisting decision making related to RUL identification. DT technology involves creating a digital replica of physical assets, capturing real-time data to mirror their operational behavior (*Abdullahi, Longo & Samie, 2024*). This process includes data acquisition from sensors and IoT devices, data integration to form a cohesive dataset, model development using simulation tools and ML algorithms, continuous monitoring, and fault identification and prediction through ML analysis (*Alam & El Saddik, 2017*; *Solari, Lysova & Montanari, 2023*).

While, virtual assistants provide interactive support and decision-making capabilities, aiding maintenance teams in managing RUL and detecting anomalies. VA technology uses AI to process data, interact with users through conversational interfaces and implement automated actions based on predictive analytics thus to predict RUL estimation. VA tools are categorized into chatbots, virtual advisors, and autonomous agents, each providing varying degrees of interaction and decision-making capabilities. Research indicates significant benefits of integrating DT and VA with ML for predictive maintenance. *Lu & Li (2023)* show that for rolling element bearings, a hybrid DT and LSTM model significantly improved RUL prediction accuracy by integrating simulation data with experimental data, achieving over 97.5% accuracy.

## Challenges

Determining the RUL of equipment is a complex issue, without a standardized approach, numerous published works from various perspectives, each offering new insights and findings on maintenance strategies and performance monitoring, highlighting different strengths and weaknesses (*Mehta, Prabhu Bam & Prabu Gaonkar, 2024*). While integration with ML to model a prediction task requires comprehensive historical machine defect, sequences of reactive action and different kind of failure data to build a robust dataset. This often results in generation of a huge amount of processing data which requires a special infrastructure and expert knowledge (*Rozhkovskaya & Sychev, 2020*).

In addition, most institution are reluctant to share their data publicly due to concerns over privacy and competition issue. Despite various methods implemented in predicting RUL, there remains significant potential for improvement and optimization, particularly in healthcare and emerging markets (*Arunan et al., 2023*). The application of these advanced algorithms to predict RUL in upgraded equipment could significantly enhance the reliability and performance of such systems, yet this remains underexplored.

The concept of midlife upgrades, which extending the life of equipment through component upgrades, has not been explored extensively. Existing studies primarily from a

theoretical perspective, with limited empirical evidence on its effectiveness (*Khan, West & Wuest, 2020*; *Wang & Zhao, 2022*).

The significance of accurate RUL prediction in healthcare is heightened by the essential role of medical devices in healthcare. Equipment malfunctions can have severe repercussions, affecting patient outcomes and operational efficacy. However, maintenance procedures for medical equipment have not completely utilized the capabilities of RUL prediction models. Most existing methods depend on reactive or preventative maintenance, which are less effective than predictive approaches driven by RUL predictions. Integrating RUL prediction models into healthcare systems enables hospitals and clinics to adopt a proactive maintenance strategy, thereby decreasing downtime, lowering expenses, and maintaining continuous patient care. Medical devices, including imaging machines, ventilators, and surgical instruments, frequently function in complex environments characterized by extremely varied usage patterns. This unpredictability adds complications to the development of precise RUL models. The substantial expense of medical equipment and its essential function in diagnosis and treatment highlight the necessity for accurate and dependable RUL projections. *Khan, West & Wuest (2020)*, empirical research regarding the efficacy of midlife enhancements in healthcare is limited, highlighting an essential want for studies that integrate RUL prediction models with upgrade plans to maximize the usage of medical equipment (*Wang & Zhao, 2022*). The hesitance of healthcare organizations to disclose operational data, stemming from privacy and competitive apprehensions, exacerbates the implementation of RUL prediction models. Collaborative initiatives and anonymised data-sharing frameworks may facilitate the resolution of these obstacles, allowing researchers to create more robust and generalizable models. Moreover, integrating IoT devices and sensor data may yield real-time insights into equipment performance, hence improving the precision of RUL projections.

## CONCLUSIONS

This review establishes a critical foundation for future research aimed at improving the integration of RUL, FDD, and anomaly detection within predictive maintenance frameworks. While significant progress has been made, continuous challenges such as limited model generalizability, low interpretability, and lack of integration across heterogeneous datasets in industrial environments continue to hinder practical deployment. The analysis underscores that most current approaches treat RUL prediction and anomaly detection in isolation, missing the synergistic potential of a unified framework. Furthermore, despite the promise shown by hybrid and ensemble AI models, these methods remain underutilized in operational environments where real-time accuracy, reliability, and explainability are crucial. To address these gaps, this review advocates for a new research direction centered on the development of integrated, explainable, and adaptive AI frameworks capable of handling noisy, incomplete, or imbalanced sensor data while maintaining predictive accuracy across diverse use cases. Thus, by leveraging the strengths of deep learning, hybrid modeling, and transfer learning, and embedding them within a fault-aware decision-support system, future research can

significantly enhance the accuracy of actual lifespan predictions for equipment in complex settings. This study identifies the current limitations in AI approaches and proposes a roadmap for advancing predictive maintenance through intelligent systems that are more aligned with operational realities. The findings are expected to contribute to smarter, data driven maintenance strategies, reduced downtimes and extended asset life span.

## ACKNOWLEDGEMENTS

The authors would like to thank the Ministry of Health, Malaysia, for providing & annotating the data.

### Funding

This work was supported by the Asian Universities Alliance—The United Arab Emirates University (AUA-UAEU Joint Research Program) (No. 12R239). The funders had no role in study design, data collection and analysis, decision to publish, or preparation of the manuscript.

### Grant Disclosures

The following grant information was disclosed by the authors:
Asian Universities Alliance—The United Arab Emirates University (AUA-UAEU Joint Research Program): 12R239.

### Competing Interests

The authors declare that they have no competing interests.

### Author Contributions

- Mohd Khidir Gazali conceived and designed the experiments, performed the experiments, analyzed the data, performed the computation work, prepared figures and/or tables, authored or reviewed drafts of the article, and approved the final draft.
- Khairunnisa Hasikin conceived and designed the experiments, performed the experiments, analyzed the data, performed the computation work, authored or reviewed drafts of the article, and approved the final draft.
- Khin Wee Lai conceived and designed the experiments, authored or reviewed drafts of the article, and approved the final draft.
- Aizat Hilmi Zamzam conceived and designed the experiments, performed the experiments, analyzed the data, authored or reviewed drafts of the article, and approved the final draft.
- Rafat Damseh conceived and designed the experiments, authored or reviewed drafts of the article, and approved the final draft.

### Data Availability

This is a literature review.

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
