# Peer review of "State-of-the-art artificial intelligence approaches for anomaly detection and remaining useful life prediction: a review"

_PeerJ Computer Science, doi:10.7717/peerj-cs.3056_

## Round 0.1 · original submission · Major Revisions

The referral process is now complete. While finding your paper interesting and worthy of publication, the referees and I feel that more work could be done before the paper is published. My decision is therefore to provisionally accept your paper subject to major revisions. More details are needed. Moreover, why we need this paper is not clear.

Reviewer 1 ·

Basic reporting

No comment.

Experimental design

No comment.

Validity of the findings

No comment.

Additional comments

In this review, machine learning techniques used for anomaly detection and RUL (Remaining Useful
Life) prediction have been systematically evaluated, focusing on their effectiveness and practical
application. However, there are a few deficiencies/errors in the study. These deficiencies/errors and
suggestions are given below as items.

1) In the "Abstract" section, there is no need to specify "Background, Methodology, Results, and
Conclusion." The abstract should be written as a cohesive whole.
2) The motivation, aim, and contributions of the study can be expressed in more detail.
3) The resolution of the figures should be high, and the text within them should be legible.
4) Are there other reviews related to the topic? If so, why were they not mentioned or compared?
5) The organization of the paper is very poor. It needs to be improved.
6) Information about the types of data or datasets used in machine learning-based methods can be
provided. Additionally, performance metrics of machine learning methods, such as accuracy, should
be discussed

Annotated reviews are not available for download in order to protect the identity of reviewers who chose to remain anonymous.

Reviewer 2 ·

Basic reporting

In this article, the authors have conducted a literature review study that includes a comprehensive analysis of Anomaly Detection and Remaining Useful Life Prediction. However, there are major concerns as follows:

- The statement expressed by the authors in the sentence below is not entirely accurate. Even though deep learning may be considered a black box, it is still a method that yields highly successful results, and similarly, in machine learning methods, it is difficult to understand the reasons behind the results.
"Complex models such as deep learning networks often act as black boxes making it challenging for maintenance staff to accept the results and comprehend the reasoning behind the forecasts."

- The studies presented in Table 7 should be cited in the format [x], as applied in the rest of the article. The information regarding the year and author names can be provided in a separate column.

Experimental design

- Some items under the Research Question section already have answers that are well-known in the literature and do not present any innovative contribution—for example, RQ2, RQ5, etc. Table 2 should be revised in terms of its contribution to the literature. Additionally, all tables are labeled as Table 1, which should be corrected.

- One of the authors' mistakes is the confusion between machine learning and deep learning concepts. Although the authors state in the title that the article reviews machine learning methods, the papers they reviewed also include deep learning methods. In this context, using the term 'machine learning' in the title narrows the scope of the review. Therefore, it would be more appropriate to use 'Artificial Intelligence' as a general term.

Validity of the findings

- The 'Components Involved' heading presented in Table 10 should be improved. For example, the information given for 'Method CNN' is quite vague. What differentiates CNN architectures is not just whether they use convolution layers, activation functions, or pooling layers, but how they design the architecture when using these components. For instance, some CNN architectures apply multi-scale feature extraction using layers with different kernel sizes in the convolution layers.

- The authors should emphasize how this review differs from similar ones in the literature. What research gaps does it fill, and what benefits does it provide for future studies? This section should be stronger and offer clearer guidance to the reader.

Reviewer 3 ·

Basic reporting

This paper reviews the effectiveness of current machine learning approaches for fault detection and predicting the RUL of equipment.

- The article generally contains grammatical errors and needs revisions in terms of language and expression.
- Adding 'deep learning' to the keywords used by the authors in their search will improve the scope and impact of the current article in the literature. This way, examples of deep learning methods, which have been a popular topic in recent years, can be included in the relevant field, and it will attract more attention from readers.
- It is important to address the strengths and weaknesses of the reviewed papers. This will provide researchers with insights for future studies.
- The detailed information and the success provided by the studies presented in Table 7, along with their contribution to the literature, can be summarized in a few sentences. This would help better understand the relevance and quality of these studies in the literature.
- Although the study systematically covers machine learning methods, a more investigation into how these models are applied in different sectors could be conducted. Currently, the article appears to focus more on theoretical aspects.
- More examples and discussions on how machine learning methods are used in different industries should be added. In addition to theoretical discussions, there should be a focus on how these methods provide practical benefits.
- The process of applying PRISMA criteria should be explained in more detail. This will demonstrate that the systematic literature review is based on a solid foundation.
- There is a need for more real-world examples and scenarios.
- Some research questions are unnecessary. For example, RQ2 and RQ5. Certain research questions, such as RQ8 and RQ9, could be combined.

Experimental design

-

Validity of the findings

-

Additional comments

-

---

## Round 0.2 · Major Revisions

The review process is now complete. While finding your paper interesting, the referees and I feel that more work could be done before the paper is published. My decision is therefore to provisionally accept your paper subject to major revisions.

Reviewer 1 ·

Basic reporting

No comment.

Experimental design

No comment.

Validity of the findings

No comment.

Additional comments

It can be observed that the study has undergone several updates based on the points mentioned in the previous review. However, it seems that some issues have not been adequately addressed or considered. These issues are as follows:
1) The purpose, motivation, and contributions of the study should be clearly and comprehensively stated. These can be presented as separate subsection.
2) The text within some figures is not legible. The resolution of the figures should be adjusted accordingly.
3) Related studies can be discussed in a separate subsection. Additionally, the differences between this study and related studies can be presented in a comparative table.
4) The organization of the paper has not been sufficiently improved.
5) Detailed information about the datasets should be provided.

Reviewer 2 ·

Basic reporting

The authors have completed the revision process successfully.

Experimental design

The authors have completed the revision process successfully.

Validity of the findings

The authors have completed the revision process successfully.

Reviewer 3 ·

Basic reporting

The authors have made the necessary revisions.

Experimental design

-

Validity of the findings

-

---

## Round 0.3 · accepted · Accept

We are happy to inform you that your manuscript has been accepted after a minor revision. Please address the comments. (Resolution of the figures)

Reviewer 1 ·

Basic reporting

No comment.

Experimental design

No comment.

Validity of the findings

No comment.

Additional comments

Based on the points mentioned in the previous review, it was observed that the study has undergone significant updating. However, the problem with the resolution of the figures remains.